# Transferable Boltzmann Generators

**Leon Klein**
Freie Universität Berlin
leon.klein@fu-berlin.de

**Frank Noé**
Microsoft Research AI4Science
Freie Universität Berlin
Rice University
franknoe@microsoft.com

## Abstract

The generation of equilibrium samples of molecular systems has been a long-standing problem in statistical physics. Boltzmann Generators are a generative machine learning method that addresses this issue by learning a transformation via a normalizing flow from a simple prior distribution to the target Boltzmann distribution of interest. Recently, flow matching has been employed to train Boltzmann Generators for small molecular systems in Cartesian coordinates. We extend this work and propose a first framework for Boltzmann Generators that are transferable across chemical space, such that they predict zero-shot Boltzmann distributions for test molecules without being retrained for these systems. These transferable Boltzmann Generators allow approximate sampling from the target distribution of unseen systems, as well as efficient reweighting to the target Boltzmann distribution. The transferability of the proposed framework is evaluated on dipeptides, where we show that it generalizes efficiently to unseen systems. Furthermore, we demonstrate that our proposed architecture enhances the efficiency of Boltzmann Generators trained on single molecular systems.

## 1 Introduction

Generative models have demonstrated remarkable success in the physical sciences, including protein structure prediction [1, 2, 3], generation of de novo molecules [4, 5, 6, 7], and efficiently generating samples from the Boltzmann distribution [8, 9, 10]. In this work, we will focus on the latter for molecular systems, which represents a promising avenue for addressing the sampling problem. The sampling problem refers to the long-standing challenge in statistical physics to generate samples from equilibrium Boltzmann distributions $\mu(x) \propto \exp\left(-U(x)/k_B T\right)$, where $U(x)$ is the potential energy of the system, $k_B$ the Boltzmann constant, and $T$ the temperature. Traditionally, samples are generated with sequential sampling algorithms such as Markov Chain Monte Carlo and Molecular Dynamics (MD) simulations. However, these algorithms require a significant amount of time to generate uncorrelated samples from the target distribution. This is due to the necessity of performing small update steps, in the order of femtoseconds, for stability. This is especially challenging in the presence of well-separated metastable states, where transitions are unlikely due to high energy barriers. In recent years, numerous machine learning methods have emerged to address this challenge [8, 11]. One such method is the Boltzmann Generators (BG) [8]. In this work, we refer to BGs as a model that allows for the approximate sampling of the Boltzmann distribution of interest and the subsequent reweighting to the unbiased target distribution. If the model is only capable of generating approximate samples, which may stem from a subset of the Boltzmann distribution, we refer to them as Boltzmann Emulators[1]. Boltzmann Generators transform a typically simple prior distribution into an approximation of the target Boltzmann distribution through a normalizing flow [12, 13, 14, 15]. Once generated, samples can be reweighted to align with the unbiased target

---

[1]To the best of our knowledge Bowen Jing introduced the name first.

distribution. The effectiveness of this reweighting hinges on how closely the generated distribution approximates the target. As a result, it is possible to obtain uncorrelated and unbiased samples from the target Boltzmann distribution, potentially achieving significant speed-ups compared to classical MD simulations.

There are numerous ways to construct a Boltzmann Generator due to the variety of realizations of normalizing flows available. In this work, we concentrate on continuous normalizing flows (CNFs) [16, 17], as opposed to coupling flows [18]. Recently, flow matching [19, 20, 21, 22] has emerged as an alternative training method for CNFs. This approach is simulation-free, enabling more efficient training of CNFs.

Thus far, Boltzmann Generators have been limited by the requirement to train them on the specific system of interest. This training process demands a significant amount of time, making it difficult to achieve substantial speed-ups over classical MD simulations. Consequently, there is a strong desire for a transferable Boltzmann Generator that can be trained on one set of molecules and effectively generalize to another, enabling efficient generation of Boltzmann samples at inference time without the need for retraining. Recently, reliable Boltzmann Generators in Cartesian coordinates for molecules have been introduced [23, 24], paving the way for transferable Boltzmann Generators. This advancement is particularly significant because these models do not rely on molecule-specific internal coordinate representations, which have traditionally made the construction of transferable models challenging.

In this work, we present a framework for *transferable* Boltzmann Generators based on CNFs, enabling effective sample generation from previously unseen Boltzmann distributions. Transferable Boltzmann Generators are particularly advantageous as they do not require retraining for similar systems and can be trained on shorter trajectories that may not fully capture all metastable states.

We make the following main contributions:

1. To the best of our knowledge, we introduce the first *transferable* Boltzmann Generator. We demonstrate its transferability on dipeptides, successfully generating unbiased samples from the Boltzmann distributions of unseen dipeptides.
2. We outline a general framework for training and sampling with transferable Boltzmann Generators based on continuous normalizing flows, which also includes the post-processing of generated samples.
3. We conduct several ablation studies to investigate the effects of different architectures, training set sizes, and biasing of the training data. The results reveal that even small training sets can suffice to train transferable Boltzmann Generators. Additionally, we compare our model with Timewarp [11], which employs large time steps instead of generating independent samples as our method does.

## 2   Related work

The initial work on Boltzmann Generators [8] has led to a great deal of subsequent research. The most common application of BGs is to generate samples from Boltzmann distributions of molecules [25, 26, 27, 28, 29, 30], as well as lattice systems [26, 31, 32, 33]. Most BGs and related methods for molecular systems require system-specific featurizations such as internal coordinates [8, 34, 26, 27, 35, 36, 30, 37, 38]. Only recently, BGs for small molecular systems in Cartesian coordinates were introduced [23, 24], using CNFs and coupling flows, respectively. Equivariant normalizing flows [39, 40, 41, 28, 23, 42] played a pivotal role in the success of Boltzmann Generators in Cartesian coordinates, not only for molecular systems. The majority of BGs employ a Gaussian prior distribution, but it is also possible to start from prior distributions close to the target distribution [43, 36, 44], which makes the learning task simpler. However, all previous Boltzmann Generators are not transferable. Arguably, the work of [7, 45] represents an exception, as they are able to generate samples from unseen conditional (Boltzmann) distributions in torsion space. However, the distribution is conditioned on a single local structure for each molecule, namely fixed bonds and angles. Consequently, in contrast to our work, they are unable to generate samples from the full Boltzmann distribution in Euclidean space. The first transferable deep generative model that was able to generate asymptotically unbiased samples from the Boltzmann distribution is [11]. Instead of generating independent samples, they learn large time steps and combine these with Metropolis-

Hastings acceptance steps, to ensure asymptotically unbiased samples. However, in contrast to our work, they do not generate uncorrelated samples.

Boltzmann Emulators are analogous to Boltzmann Generators, yet they are not designed to generate unbiased equilibrium samples from the target Boltzmann distribution. Instead, they are intended to generate approximate samples that do not undergo reweighting. Furthermore, the generation of all metastable states may not be a necessary requirement, depending on the system. Boltzmann Emulators do not need to be based on flow models, as they do not aim to do reweighing to the target distribution. They are often similar to Boltzmann Generators and use normalizing flows or diffusion models for the architecture, but due to removing the constraint of sampling the unbiased Boltzmann distribution, they can target significantly larger systems or are transferable. One example is [46], who propose a three stage transferable CNF model to learn peptide ensembles. [47] use flow matching to learn distributions of proteins, while [48] utilize diffusion models. [49] build a transferable Boltzmann Emulator for small molecules. Others aim to additionally also capture the correct dynamics of the molecular systems, such as [50], who use a diffusion model to predict transition probabilities. Scaling to larger systems often requires coarse graining [51, 52, 47], e.g. describing amino acids by a single bead rather than the individual atoms. However, this approach precludes the possibility of reweighting to the Boltzmann distribution.

A distinct, though related, learning objective is to generate novel molecular conformations. However, approximations from the Boltzmann distribution are not necessary; it is sufficient to generate a few (or even a single) conformation per molecule. The utilized architectures are once again analogous, as flow and diffusion models are employed [5, 6, 53, 54, 55, 7].

# 3 Boltzmann Generators and Normalizing Flows

Here, we describe Boltzmann Generators and normalizing flows, which are a central part of our proposed transferable Boltzmann Generator framework. We follow the notation of [23].

## 3.1 Boltzmann Generators

Boltzmann Generators (BGs) [8] combine an exact likelihood deep generative model and a reweighting algorithm to reweight the generated distribution to the target Boltzmann distribution. The exact likelihood deep generative model is trained to generate samples from a distribution $\tilde{p}(x)$ that is close to the target Boltzmann distribution $\mu(x)$. A common choice for the exact likelihood model are normalizing flows.

The Boltzmann Generator can be used to generate unbiased samples by first sampling $x \sim \tilde{p}(x)$ with the exact likelihood model and then computing corresponding importance weights $w(x) = \mu(x)/\tilde{p}(x)$ for each sample. These allow to reweight generated samples to the target Boltzmann distribution $\mu(x)$. It is possible to estimate observables of interest (asymptotically unbiased) using the weights $w(x)$ with importance sampling via

$$\langle O \rangle_\mu = \frac{\mathbb{E}_{x \sim \tilde{p}(x)}[w(x)O(x)]}{\mathbb{E}_{x \sim \tilde{p}(x)}[w(x)]}. \tag{1}$$

Furthermore, these reweighting weights can be employed to assess the efficiency of trained BGs by computing the effective sample size (ESS) with Kish's equation [56]. In this work, we will compute the relative ESS, rather than the absolute one, and refer to it as ESS.

## 3.2 Continuous Normalizing Flows (CNFs)

Normalizing flows [15, 57] are a type of deep generative model used to learn complex probability densities $\mu(x)$ by transforming a prior distribution $q(x)$ through an invertible transformation $f_\theta : \mathbb{R}^n \to \mathbb{R}^n$, resulting in the push-forward distribution $\tilde{p}(x)$.

Continuous Normalizing Flows (CNFs) [16, 17] are a specific kind of normalizing flow. For CNFs, the invertible transformation $f_\theta^t(x)$ is defined by the ordinary differential equation

$$\frac{df_\theta^t(x)}{dt} = v_\theta\left(t, f_\theta^t(x)\right), \quad f_\theta^0(x) = x_0, \tag{2}$$

where $v_\theta(t, x) : \mathbb{R}^n \times [0, 1] \to \mathbb{R}^n$ is a time-dependent vector field. The solution to this initial value problem provides the transformation equation

$$f_\theta^t(x) = x_0 + \int_0^t dt' v_\theta\left(t', f_\theta^{t'}(x)\right),\tag{3}$$

with $f_\theta^1(x) = \tilde{p}^t(x)$. The corresponding change in log density from the prior to the push-forward distribution is described by the continuous change of variable equation

$$\log \tilde{p}(x) = \log q(x) - \int_0^1 dt \nabla \cdot v_\theta\left(t, f_\theta^t(x)\right).\tag{4}$$

**Equivariant flows** The energy of molecular systems is typically invariant under permutations of interchangeable particles and global rotations and translations. Consequently, it is advantageous for the push-forward distribution of a Boltzmann Generator to possess the same symmetries as the target system. In [39, 40] the authors demonstrate that the push-forward distribution $\tilde{p}(x)$ of a permutation and rotation equivariant normalizing flow with a permutation and rotation invariant prior distribution, is again rotation and permutation invariant. Furthermore, [39] present a method to construct such equivariant CNFs by using an equivariant vector field $v_\theta$.

### 3.3 Flow matching

Flow matching [19, 20, 21, 22] enables efficient, simulation-free training of CNFs. The conditional flow matching training objective allows for the direct training of the vector field $v_\theta(t, x)$ through

$$\mathcal{L}_{\mathrm{CFM}}(\theta) = \mathbb{E}_{t \sim [0,1], x \sim p_t(x|z)} \left|\left|v_\theta(t, x) - u_t(x|z)\right|\right|_2^2.\tag{5}$$

There are many possible parametrizations for the conditional vector field $u_t(x|z)$ and the conditional probability path $p_t(x|z)$. One of the most simple, but powerful possible parametrization is

$$z = (x_0, x_1) \quad \text{and} \quad p(z) = q(x_0)\mu(x_1)\tag{6}$$

$$u_t(x|z) = x_1 - x_0 \quad \text{and} \quad p_t(x|z) = \mathcal{N}(x|t \cdot x_1 + (1-t) \cdot x_0, \sigma^2),\tag{7}$$

which we use in this work to train our models. For a more detailed description refer to [19, 22, 23, 53].

## 4 Transferable Boltzmann Generators

This section presents our proposed framework for transferable Boltzmann Generators (TBGs).

### 4.1 Architecture

Our proposed transferable Boltzmann Generator is based on a CNF. The corresponding vector field $v_\theta(t, x)$ is parametrized by an $O(D)$- and $S(N)$-equivariant graph neural network (EGNN) [41, 58, 46], as commonly used in prior work, e.g. [23, 41]. Although, less expressive than other equivariant networks such as [59, 60, 61, 62], it is faster to evaluate, which is important for CNFs as there can be hundreds of vector field calls during inference.

The vector field $v_\theta(t, x)$ consists of $L$ consecutive layers. The position of the $i$-th particle $x_i$ is updated according to the following equations:

$$h_i^0 = (t, a_i, b_i, c_i), \quad m_{ij}^l = \phi_e\left(h_i^l, h_j^l, d_{ij}^2\right),\tag{8}$$

$$x_i^{l+1} = x_i^l + \sum_{j \neq i} \frac{(x_i^l - x_j^l)}{d_{ij} + 1} \phi_d(m_{ij}^l),\tag{9}$$

$$h_i^{l+1} = \phi_h\left(h_i^l, m_i^l\right), \quad m_i^l = \sum_{j \neq i} \phi_m(m_{ij}^l) m_{ij}^l,\tag{10}$$

$$v_\theta(t, x^0)_i = x_i^L - x_i^0 - \frac{1}{N} \sum_j^N (x_j^L - x_j^0),\tag{11}$$

where $\phi_\alpha$ represents different neural networks, $d_{ij}$ is the Euclidean distance between particle $i$ and $j$, $t$ is the time, $a_i$ is an embedding for the particle type, $b_i$ for the amino acid, and $c_i$ or the amino acid position in the peptide. In the final step, the geometric center is subtracted to ensure that the center of positions is conserved. When combined with a symmetric mean-free prior distribution, the push-forward distribution of the CNF will be $O(D)$- and $S(N)$-invariant, as demonstrated in [63].

The embedding of each atom is constructed from three parts. The first part is the atom type $a_i$, which is a one-hot vector of $54$ classes. The classes are defined based on the atom types in the peptide topology. Therefore, only a few atoms are indistinguishable, such as hydrogen atoms that are bound to the same carbon or nitrogen atom. The second part is the amino acid to which the atom belongs, which is divided into 20 classes. The third part is the position of the amino acid in the peptide sequence. This embedding is similar to the embedding used in [46] for the rotamer embeddings. The amino acid and positional embeddings are only used for the transferable experiments. For more details see Appendix B.5. In this study, we refer to this transferable Boltzmann Generator architecture as *TBG + full*, and we use this name even when we apply it to a non-transferable setting.

The proposed architecture in [23] uses distinct encodings for all backbone atoms and the atom types for all other atoms. This represents a special case of our architecture, wherein $b_i$ and $c_i$ are omitted and $a_i$ encodes the atom type or a backbone atom. Hence, there are 13 classes for $a_i$. We refer to this architecture as *TBG + backbone*. Furthermore, we refer to the specific architecture employed in [23] as BG + backbone for the alanine dipeptide experiments. Note that using only $a_i$ causes problems for transferability, see Appendix A.4 for more details.

Moreover, we employ a model that utilizes the atom type as the sole encoding (there are only five distinct atom types). This model is referred to as simply *TBG*.

## 4.2 Training transferable Boltzmann Generators

All transferable Boltzmann Generators utilize flow matching for training. Given the variation in peptides across batches, the flow matching loss for each peptide is normalized by the number of atoms it contains. This training procedure is applied to all different architectures. For further details, refer to Appendix B.

## 4.3 Inference with transferable Boltzmann Generators

Sampling with a transferable Boltzmann Generator, especially on unseen peptides, poses multiple challenges: (i) Some generated samples may not correspond to the molecule of interest, but rather to a molecule that contains the same atoms but has a different bonding graph. Some of these configurations might even be valid molecules. For some examples see Appendix A.4. However, as we are in this work interested in sampling from the equilibrium Boltzmann distribution for a given molecular bonding graph, rather than sampling distinct molecules, we would like to avoid these cases. Nevertheless, this effect can be largely mitigated by our proposed TBG + full architecture. (ii) When working with classical force fields, the correct ordering with respect to the topology is crucial for evaluating energies. This is not a concern for semi-empirical force fields, as they respect the permutation symmetry of particles of the same type. As we typically use a Gaussian prior distribution, it is common that the generated samples are not arranged according to the topology. Consequently, in order to evaluate the energy, it is necessary to reorder the generated samples according to the topology. (iii) It is possible that the chirality of generated samples differs from that of the peptide of interest.

We resolve (i) and (ii) by generating a bond graph for the generated samples, based on empirical bond distances and atom types. This graph is then compared with a reference bond graph. If the two graphs are isomorphic, we can conclude that the configuration is correct. For more details, see Appendix B.1. For (iii), we employ the code of [11] to check all chiral centers. If all chirality centers of a peptide are flipped, this can be resolved by mirroring. Otherwise, these samples are assigned high energies, as they are not from the target Boltzmann distribution of interest. It is important to note that only generated samples with the correct configuration and chirality are considered valid samples from the Boltzmann distribution of interest.

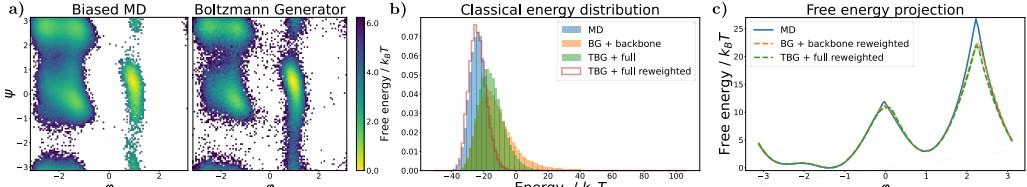

Figure 1: Results for the alanine dipeptide system simulated with a classical force field (a) Ramachandran plots for the biased MD distribution (left) and for samples generate with the TBG + full model (right). (b) Energies of samples generated with different methods. (c) Free energy projection along the slowest transition ($\varphi$ angle), computed with different methods.

Table 1: Comparison of Boltzmann Generators with different architectures for the single molecular system alanine dipeptide. Errors are computed over five runs. The results for the Boltzmann Generator and backbone encoding (BG + backbone) for the semi-empirical force field are taken from [23].

| Model | NLL ($\downarrow$) | ESS ($\uparrow$) |
|---|---|---|
| | Alanine dipeptide - semi-empirical force field | |
| BG + backbone [23] | $-107.56 \pm 0.09$ | $0.50 \pm 0.13\%$ |
| TBG + full (ours) | $\mathbf{-124.71 \pm 0.08}$ | $\mathbf{1.03 \pm 0.17}\%$ |
| | Alanine dipeptide - classical force field | |
| BG + backbone [23] | $-109.02 \pm 0.01$ | $1.56 \pm 0.30\%$ |
| TBG + full (ours) | $\mathbf{-127.06 \pm 0.12}$ | $\mathbf{6.03 \pm 1.34}\%$ |

## 5 Experiments

In this section, we compare our model with similar previous work on equivariant Boltzmann Generators [23] for alanine dipeptide. Moreover, we show the transferability of our model on dipeptides, where we compare our model with the transferable Timewarp model [11]. More experimental details, such as dataset details, the specifics of the employed models, and the utilized computing infrastructure can be found in Appendix B.

### 5.1 Alanine dipeptide

In our first experiment, we investigate the single molecule alanine dipeptide in implicit solvent, described in Cartesian coordinates. The dataset was introduced in [23], for more details see Appendix B.2. The training trajectory was generated by sampling with respect to a classical force field, and subsequently, $10^5$ random samples were relaxed with respect to the semi-empirical *GFN2-xTB* force-field [64] for 100fs each. The objective is to train a Boltzmann Generator capable of sampling from the equilibrium Boltzmann distribution defined by the semi-empirical *GFN2-xTB* force-field efficiently and to recover the free energy surface along the slowest transition, i.e. the $\varphi$ angle. Following the methodology outlined in [23], the training data is biased towards the less probable (positive) $\varphi$ state. It is evident that any trained model on this set will be biased in comparison to the true Boltzmann distribution defined by the semi-empirical energy. However, the reweighting technique allows for the debiasing of the samples. The model is trained in the same way as described in [23]. Overall, the likelihoods and ESS values observed for the TGB + full model are superior to those reported in [23] (Table 1). This is achieved with nearly the same amount of parameters and maintaining comparable training and inference times (see Appendix B.3). Furthermore, the correct free energy difference is recovered, as demonstrated in Appendix A.1.

In [23] the authors utilized a semi-empirical potential to avoid the required ordering of the atoms to the topology for classical force fields. As the prior distribution of the Boltzmann Generator is usually a multivariate standard Gaussian distribution, generated samples will almost certainly not have the correct ordering. As we have introduced an efficient way to reorder samples in Section 4.3, we can now also evaluate alanine dipeptide for a classical force field. Therefore, we retrain the model in [23] on the classical MD trajectory and compare with our TBG + full architecture. We bias the training

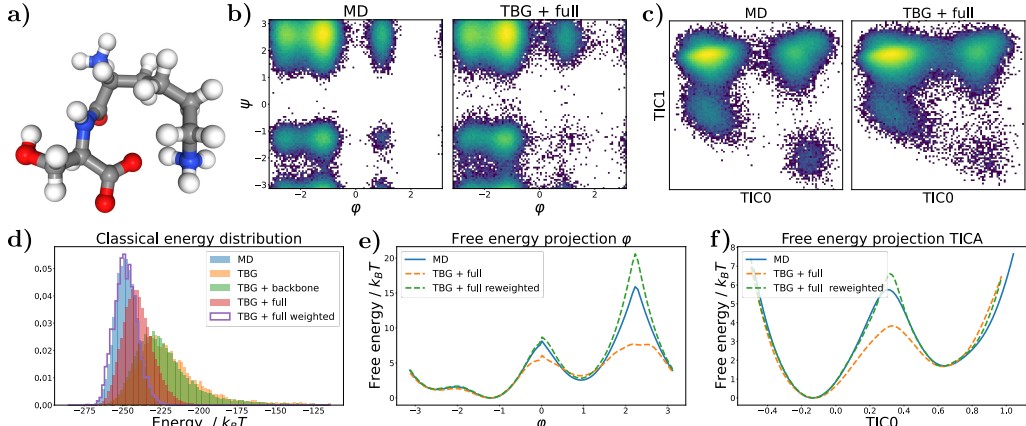

Figure 2: Results for the KS dipeptide (a) Sample generated with the TBG + full model (b) Ramachandran plot for the weighted MD distribution (left) and for samples generate with the TBG + full model (right). (c) TICA plot for the weighted MD distribution (left) and for samples generate with the TBG + full model (right). (d) Energies of samples generated with different methods and architectures. (e) Free energy projection along the $\varphi$ angle. (f) Free energy projection along the slowest transition (TIC0).

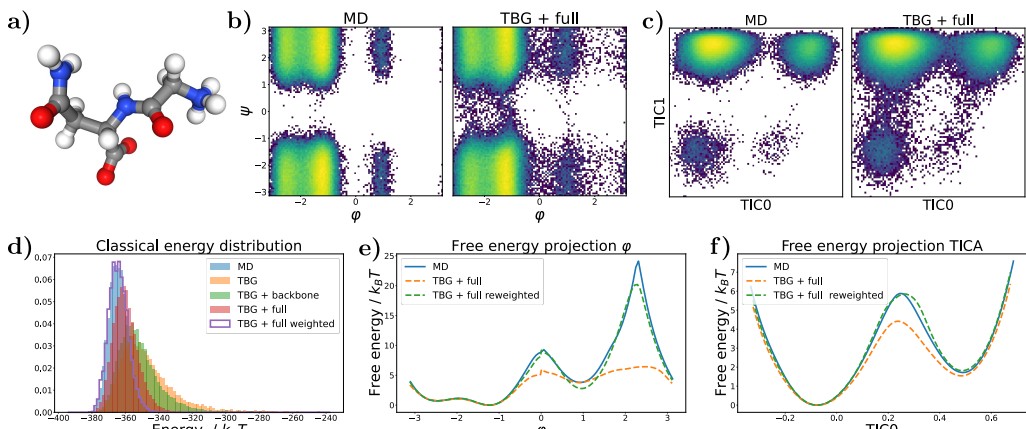

Figure 3: Results for the GN dipeptide (a) Sample generated with the TBG + full model (b) Ramachandran plot for the weighted MD distribution (left) and for samples generate with the TBG + full model (right). (c) TICA plot for the weighted MD distribution (left) and for samples generate with the TBG + full model (right). (d) Energies of samples generated with different methods and architectures. (e) Free energy projection along the $\varphi$ angle. (f) Free energy projection along the slowest transition (TIC0).

data as before towards the unlikely $\varphi$ state. As expected, the likelihood and ESS for the classical force field are much better than for the semi-empirical one, as the training data stems from the target distribution. Our proposed architecture again performs significantly better, as shown in Table 1 and Figure 1. The majority of generated samples with the TBG + full model and the BG + backbone sample nearly exclusively correct configurations, i.e. configurations with the correct bond graph, namely nearly $100\%$ and about $98\%$, respectively. As presented in Figure 1, both models recover the free energy landscape correctly.

## 5.2 Dipeptides (2AA)

In our second experiment, we evaluate our model on dipeptides and show transferability. The dataset was introduced in [11]. The training set consists of 200 dipeptides, which were simulated each with a

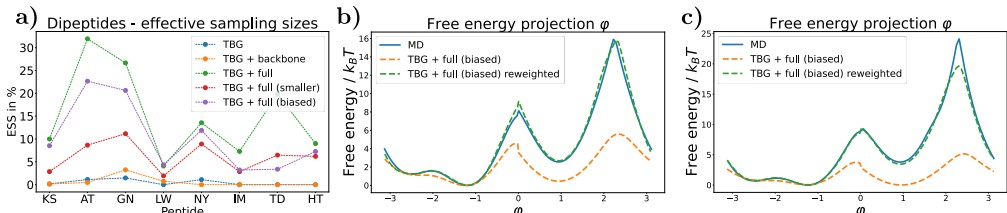

Figure 4: (a) Effective samples sizes (ESS) for the first 8 test peptides for different transferable architectures and training sets. (b) Free energy projection along the $\varphi$ angle for the TBG + full model trained on the *biased* dataset for the KS dipeptide. The weighted free energy projection demonstrates a superior fit compared to the TBG + full model (see Figure 2e). (c) Free energy projection along the $\varphi$ angle for the TBG + full model trained on the *biased* dataset for the GN dipeptide. The weighted free energy projection demonstrates a superior fit compared to the TBG + full model (see Figure 3e).

classical force field for 50 ns and, therefore, may not have reached convergence. Nevertheless, as previously demonstrated, it is not necessary to train on unbiased data in order to obtain unbiased samples with a Boltzmann Generator.

We compare the three different transferable architectures described in Section 4.1 and use the same training procedure for all of them. Similar to the alanine dipeptide experiments, we obtain significantly better results for the TBG + full model in terms of ESS (Table 2 and Figure 4a), energies (Figure 2d), the ratio of correct configurations (Table 2), and likelihoods of test set samples (Appendix A.5). In particular, the extremely low number of correct configurations for numerous test peptides for the TBG and TBG + backbone models renders them unsuitable as Boltzmann Generators for this setting (Table 2 and Appendix A.5). Furthermore, the TBG + full model always finds all metastable states for unseen test peptides (see also Appendix A.5).

The results for the well-performing TBG + full model are presented for two example peptides from the test set in Figure 2 and Figure 3. They were chosen as all architectures sample relevant amounts of valid configurations. Detailed results for other evaluated test peptides are shown in Appendix A.5.

The TBG + full model is an exemplary Boltzmann Emulator, as it is capable of capturing all metastable states of the target Boltzmann distribution (Figure 2b,c and Figure 3b,c). However, it is furthermore also a capable Boltzmann Generator, as it allows for efficient reweighting (Figure 2d,e,f and Figure 3d,e,f) with good ESSs (Table 2). To identify different metastable states, we employ time-lagged independent component analysis (TICA) [65], a dimensionality reduction technique that separates metastable states. We show this analysis in addition to the Ramachandran plots for the dihedral angles.

Moreover, we investigate the influence of the training set in two ablation studies.

**Training on a biased training set**  Our alanine dipeptide results as well as [23] indicate that it can be advantageous to bias the training data towards states that are less probable, such as positive $\varphi$ states, to recover free energy landscapes. Therefore, we bias the training data by weighting positive $\varphi$ states for each training peptide, such that they have nearly equal weight to the negative states (see also Appendix B.4). We show that a TBG + full model trained on this dataset (TBG + full (biased)) produces even more accurate free energy landscapes for both the Ramachandran and TICA projections (Figure 4bc). Notably, the unweighted projection shows a clear bias, as expected. However, as the training data is now biased, the effective sample size (ESS) is generally lower (Table 2 and Figure 4a).

**Training on a smaller training set**  Additionally, we examine the impact of smaller training sets on the generalization results. To this end, we train the TBG + full model on two smaller datasets with shorter simulation times: (i) 5ns for each training simulation and (ii) only 500ps of each training simulation. Consequently, the training trajectories are 10 times and 100 times smaller than before. As we utilize only the initial portion of each trajectory, a greater number of metastable states are missed during the brief simulations, as illustrated in Appendix A.2. Training on the tenfold smaller trainings set, we refer to the model as TBG + full (smaller), shows slightly worse results compared to training on the whole trainings set (Table 2 and Appendix A.5). The even smaller trainings set leads to inferior results, with several metastable states being missed as presented inAppendix A.3.

Table 2: Effective samples size and correct configuration rate for unseen dipeptides across different transferable Boltzmann Generator (TBG) architectures. The values are computed for 8 test dipeptides. See Appendix A.5 for more results.

| Model | ESS (↑) | | Correct configurations (↑) | |
| --- | --- | --- | --- | --- |
| | Mean | Range | Mean | Range |
| TBG | $0.48 \pm 0.59\%$ | $(0.0\%, 1.47\%)$ | $13 \pm 18\%$ | $(1\%, 48\%)$ |
| TBG + backbone | $0.58 \pm 1.04\%$ | $(0.0\%, 3.24\%)$ | $17 \pm 21\%$ | $(1\%, 52\%)$ |
| TBG + full | $\mathbf{15.29 \pm 9.27}\%$ | $(\mathbf{4.08}\%, \mathbf{31.93}\%)$ | $\mathbf{98 \pm 2}\%$ | $(\mathbf{94}\%, \mathbf{100}\%)$ |
| TBG + full (smaller) | $6.13 \pm 3.13\%$ | $(1.93\%, 11.16\%)$ | $96 \pm 3\%$ | $(88\%, 100\%)$ |
| TBG + full (biased) | $\mathbf{10.24 \pm 7.14}\%$ | $(\mathbf{3.21}\%, \mathbf{22.66}\%)$ | $\mathbf{98 \pm 2}\%$ | $(\mathbf{93}\%, \mathbf{100}\%)$ |

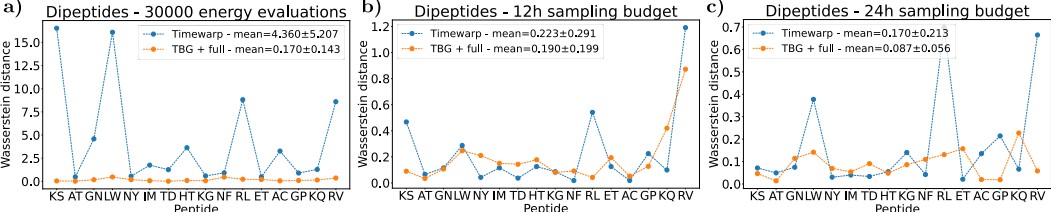

Figure 5: TBG / Timewarp MCMC [23] sampling experiments. Wasserstein distance between the generated Ramachandran plot and the MD Ramachandran plot for different computational budgets for dipeptides. Lower is better. (a) After 30000 energy evaluations (b) After 12h wall-clock-time. (c) After 24h wall-clock-time.

Nevertheless, these findings show that transferable Boltzmann Generators can be effectively trained using very small datasets, even when individual trajectories lack metastable states.

### 5.3 Comparison with Timewarp [11]

In this section, we compare our TBG model with the Timewarp model [11] on the dipeptide dataset. Unlike our approach, which generates independent samples, the Timewarp model predicts large time steps, which can be cobined with Metropolis-Hastings acceptance steps to ensure asymptotically unbiased sampling. Additionally, Timewarp employs a coupling flow rather than a continuous flow. Similar to our TBG + full model, the Timewarp model also treats most particles as distinguishable. It does so by conditioning the learned probability on the current state, implicitly capturing the topology information.

We compare the Wasserstein distance between the generated Ramachandran plot and the MD Ramachandran plot for different computational budgets for dipeptides: (a) 30,000 energy evaluations, (b) 12 hours, and (c) 24 hours of wall-clock simulation time on an A-100 GPU. As shown in Figure 5 the TBG + full model outperformed the Timewarp model across all budgets, particularly in terms of energy evaluations. Energy evaluations are especially critical, as they become a major computational bottleneck with more complex force fields. Moreover, the unweighted samples of the TBG + full model have lower energies, and are therefore closer to the actual target Boltzmann distribution than samples generated with Timewarp without Metropolis-Hastings acceptance (exploration mode). Additional results and details can be found in Appendix A.7.

## 6 Discussion

For the first time, we demonstrated the feasibility of training *transferable* Boltzmann Generators. We introduced a general framework for training and evaluating transferable Boltzmann Generators based on continuous normalizing flows. Furthermore, we developed a transferable architecture based on equivariant graph neural networks and demonstrated the importance of including topology information in the architecture to enable efficient generalization to unseen, but similar systems. The transferable Boltzmann Generator was evaluated on dipeptides, where significant effective sample sizes were demonstrated on unseen test peptides and accurate sampling of physical properties, such

as the free energy difference between metastable states, was achieved. Moreover, we have shown in ablation studies that transferable Boltzmann Generators can be extremely data efficient, with even small training trajectories being sufficient. Future research will determine whether and how transferable Boltzmann Generators can be scaled to larger systems.

## 7    Limitations / Future work

Scaling transferable Boltzmann Generators to larger systems remains for future work. Notably, this usually requires large amounts of computational resources, as e.g. shown in [11], where they are able to train their transferable model on tetrapeptides, but use more than 100 times more parameters than us. Scaling to larger systems often involves coarsegraining, which typically results in the loss of an explicit energy function. In such cases, the transferable Boltzmann Generator would effectively become a transferable Boltzmann Emulator. Depending on the specific application, samples from a distribution close to the target Boltzmann distribution may be sufficient. Our results show that unweighted samples from the TBG + full model already closely resemble the target Boltzmann distribution.

Additionally, small systems can still be highly relevant, especially when paired with more expensive force fields, such as semi-empirical or first-principles quantum-mechanical force fields. In these scenarios energy evaluations become a primary bottleneck. Boltzmann Generators are particularly advantageous in this context, as they require several orders of magnitude fewer energy evaluations compared to MD simulations or other iterative methods, such as Timewarp [11]. One potential real world application would therefore be the simulation of small molecules with expensive force fields.

Instead of flow matching, one could use optimal transport flow matching [22] or equivariant optimal transport flow matching [23, 53] for training, but as indicated in [23] the gains for molecular systems, especially in the presence of many distinguishable particles, are small.

Throughout our work, we utilize a standard Gaussian prior distribution. However, as recently introduced, an alternative is to use a Harmonic prior distribution [66, 67], where atoms that are close in the bond graph are sampled in the vicinity of each other. Notably, We experimented with this Harmonic prior but found no significant improvements for our transferable model. This aligns with findings by [68], indicating that chemically informed priors do not enhance performance substantially compared to simpler uninformed priors for flow matching in molecular systems. Instead, the network architecture and inductive bias play a more crucial role. However, such priors might become more important for larger systems.

Despite conducting a series of ablation studies, we did not pursue the impact of a training set comprising a smaller number of peptides. Instead, we opted for investigating shorter trajectories. Another potential direction could be relaxing the 2AA dataset using a semi-empirical force field and training on this modified version, similar to the alanine dipeptide experiment. However, this approach incurs additional computational costs, as the entire 2AA dataset would need to be relaxed with respect to the semi-empirical force field.

We used the EGNN architecture for the vector field due to its fast evaluation capabilities. Future research could explore alternative architectures for the vector field, such as those proposed by [59, 60, 61, 62, 69, 70, 67], to determine if they improve performance and enable scaling to larger systems. We hope that our framework will facilitate the scaling of transferable Boltzmann Generators to larger systems in future research.

## 8    Broader Impact

This work represents foundational research with no immediate societal impact. However, if our method is scalable to larger, more relevant systems, it could facilitate the acceleration of drug and material discovery by replacing or enhancing MD simulations, which often play a crucial part in the process. A potential risk is that this method then might be used to identify new diseases or develop biological weapons. Another risk is the lack of a known convergence criterion, making it impossible to confirm that all potential configurations have been identified, even with an infinite number of samples. This could lead to false claims about the results, potentially affecting subsequent applications.

## Acknowledgements

We gratefully acknowledge support by the Deutsche Forschungsgemeinschaft (SFB1114, Projects No. C03, No. A04, and No. B08), the European Research Council (ERC CoG 772230 "ScaleCell"), the Berlin Mathematics center MATH+ (AA1-10), and the German Ministry for Education and Research (BIFOLD - Berlin Institute for the Foundations of Learning and Data). We gratefully acknowledge the computing time made available to them on the high-performance computer "Lise" at the NHR Center NHR@ZIB. This center is jointly supported by the Federal Ministry of Education and Research and the state governments participating in the NHR (`www.nhr-verein.de/unsere-partner`). We thank Andreas Krämer, Michele Invernizzi, Jonas Köhler, Atharvar Kelkar, Yaoyi Chen, Max Schebek, Iryna Zaporozhets, and Hannes Stärk for insightful discussions. Moreover, we thank Andrew Foong and Yaoyi Chen for providing the datasets from the Timewarp project [11].

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

Table 3: Dimensionless free energy differences for the slowest transition of alanine dipeptide estimated with various methods. Umbrella sampling yields a converged reference solution. Errors are calculated over five runs. Values for BG + backbone and Umbrella sampling are taken from [23].

| | Umbrella sampling | BG + backbone [23] | TBG + full (ours) |
|---|---|---|---|
| Free energy difference / $k_BT$ | $4.10 \pm 0.26$ | $4.10 \pm 0.08$ | $4.09 \pm 0.05$ |

Figure 6: Example Ramachandran plots for different trajectory lengths for the training data. It can be observed that as the trajectory length decreases, the number of metastable states that are missed increases, thereby making the learning task more challenging. (a) AY dipeptide (b) IH dipeptide.

# Appendix

## A    Additional results and experiments

### A.1    Semi-empirical force field for alanine dipeptide

We report the free energy differences for the slowest transitions of alanine dipeptide for a semi-empirical force field in Table 3. See Section 5 for more details.

### A.2    Dipeptide training data

When training on smaller training sets, i.e. with shorter trajectories, additional metastable states will not be visited during the short simulation times. We show this for two example training peptides in Figure 6. Nevertheless, the TBG + full (smaller) model trained on 10 times shorter trajectories, is nearly as good as the model trained on the full trajectories, see Section 5. However, for the 100 times smaller trajectories, the TBG + full model performs significantly worse, see Appendix A.3.

### A.3    Smaller dataset

We investigate the effect of 100 times smaller trainings trajectories, i.e. simulation time of only 500ps. As shown in Appendix A.2, these trajectories miss many metastable states. This can be also observed for the so trained models, which we refer as TBG + full (smaller500), as they do not capture especially unlikely metastable states well as presented in Figure 7 and Figure 8. In contrast, models trained on larger trajectories find all metastable states and allow for efficient reweighting, as discussed in Section 5 and Appendix A.5.

### A.4    Sampled dipeptide configurations

For some amino acid combinations, both the TBG and TBG + backbone models sample only a small number of correct configurations. Although the generated configurations are potentially valid molecular configurations, they are not the one of the target dipeptide as shown in Figure 9. This is often due to the encoding, which e.g. cannot distinguish between different orderings of amino acids in a peptide. Only the various TBG + full architectures samples nearly exclusively correct configurations.

### A.5    Additional results for dipeptides

Inference is a costly process, and extensive sampling is necessary to obtain reliable estimates for the expected sample size (ESS). Therefore, we only evaluate the transferable models on a subset of the

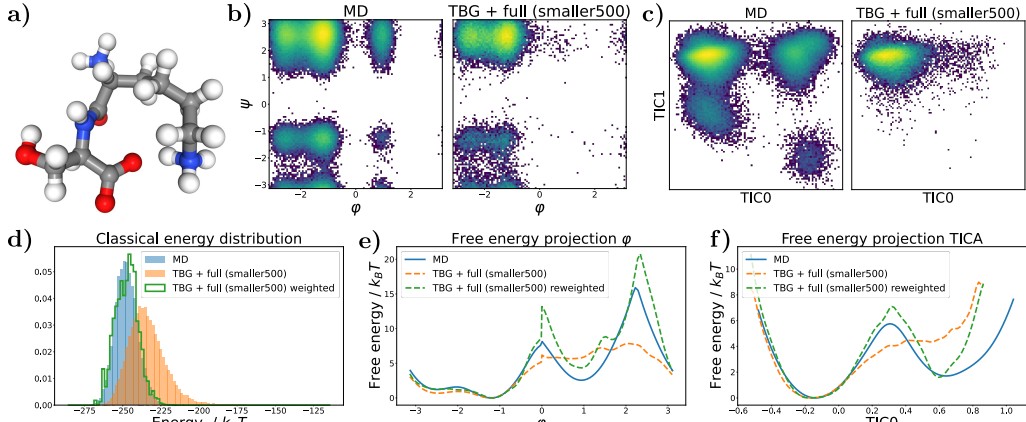

Figure 7: Results for the KS dipeptide for TBG + full model trained on 100 times smaller training trajectories. As can be seen in Figure 2, the results for the TBG + full model trained on the whole trajectories are much better. (a) KS dipeptide (b) Ramachandran plot for the weighted MD distribution (left) and for samples generate with the model (right). (c) TICA plot for the weighted MD distribution (left) and for samples generate with the model (right). (d) Energies of samples generated with the model. (e) Free energy projection along the $\varphi$ angle. (f) Free energy projection along the slowest transition (TIC0).

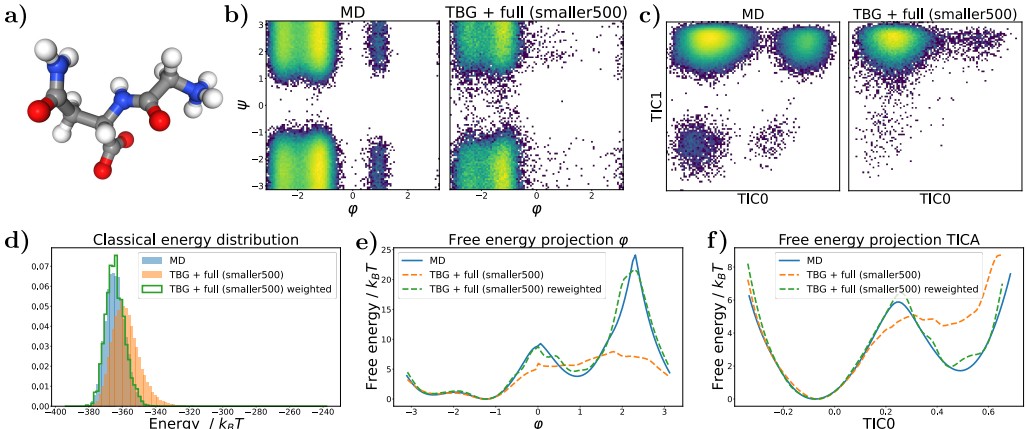

Figure 8: Results for the GN dipeptide for TBG + full model trained on 100 times smaller training trajectories. As can be seen in Figure 3, the results for the TBG + full model trained on the whole trajectories are much better. (a) GN dipeptide (b) Ramachandran plot for the weighted MD distribution (left) and for samples generate with the model (right). (c) TICA plot for the weighted MD distribution (left) and for samples generate with the model (right). (d) Energies of samples generated with the model. (e) Free energy projection along the $\varphi$ angle. (f) Free energy projection along the slowest transition (TIC0).

test set. The dipeptides are randomly selected, but it is ensured that all amino acids are represented at least once. However, we evaluate the best-performing model, namely TBG + full, for all 100 test peptides. The results for the additional test peptides are in good agreement with the first ones as presented in Table 4 and Figure 10.

We report individual results for the different architectures in Figure 10c,d.

To illustrate the performance of the TBG + full model, we present additional examples of dipeptides from the test set in Figure 11a-f and Figure 12a-f. Furthermore, we also again show results for training on the biased dataset in the same figures (Figure 11g,h,i and Figure 12g,h,i). As observed previously, the TBG + full (biased) model recovers the free energy landscape better than the TBG +

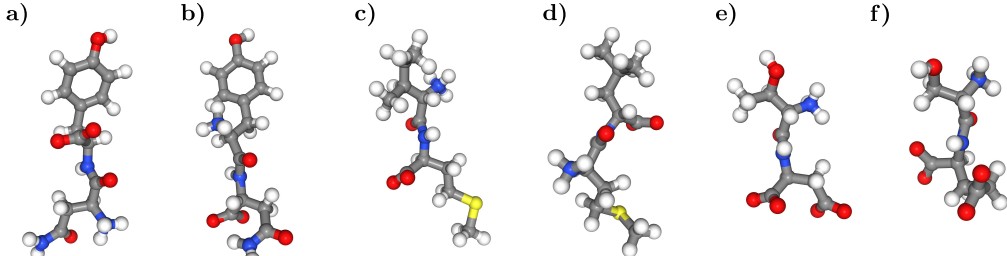

Figure 9: Sampled molecules with the TBG and TBG + backbone models, which do not have the correct topology. (a) NY dipeptide reference (b) Generated molecule with NY atoms by the TBG model. (c) IM dipeptide reference (d) Generated molecule with IM atoms by the TBG model. (e) TD dipeptide reference (f) Generated molecule with TD atoms by the TBG + backbone model.

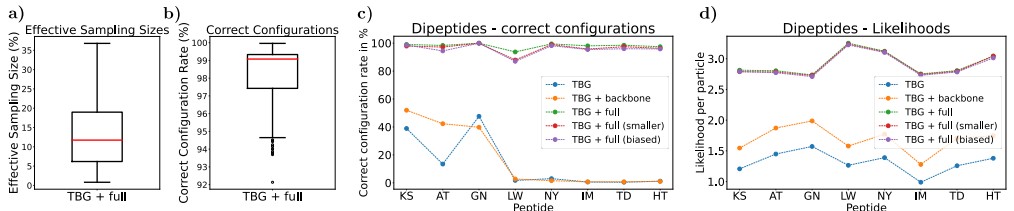

Figure 10: (a) Effective samples sizes for all 100 test dipeptides for the TBG + full model. (b) Percentage of correct configurations for all 100 test dipeptides sampled with the TBG + full model. (c,d) Performance comparison for different transferable architectures and training sets on 8 dipeptides from the test set (c) Correct configuration rate (d) Likelihood per particle.

full model, especially for the $\varphi$ projections. We present additional examples of dipeptides from the test set for the TBG + full model in Figure 13, Figure 14, Figure 15, Figure 16, and Figure 17.

Furthermore, we present results for two example peptides from the test set for the TBG + full (smaller) model, which is trained on trajectories that are tenfold smaller than those used for the TBG + full model. These results are shown in Figure 18 and Figure 19.

### A.6 Transferable Boltzmann Generators as Boltzmann Emulators

Given the high cost of sampling with CNFs, which necessitates integrating the Jacobian trace along the positions, we did not evaluate all available test peptides for all models (see Appendix B.2). However, since sampling without the Jacobian trace is less expensive and we do not require as many samples as for estimating the ESS, we also employ the TBG + full (smaller) model as a Boltzmann Emulator to ascertain whether we have identified all metastable states, despite the fact that it was only trained on the 10 times smaller training set. The Boltzmann Emulator is evaluated on a diverse set of test peptides, and nearly always finds all metastable states within less than one hour of wall clock time. This is a notable improvement over MD simulations, which often take longer to explore due to the iterative nature of MD. Some examples are shown in Figure 20. This experiment shares similarities with the exploration mode of [11], where they employ their model without the acceptance

Table 4: Effective samples size and correct configuration rate for unseen dipeptides for the TBG + full architecture for different numbers of test peptides.

| Model | ESS ($\uparrow$) | | Correct configurations ($\uparrow$) | |
| --- | --- | --- | --- | --- |
| | Mean | Range | Mean | Range |
| TBG + full (8 test peptides) | $15.29 \pm 9.27\%$ | $(4.08\%, 31.93\%)$ | $98 \pm 2\%$ | $(94\%, 100\%)$ |
| TBG + full (100 test peptides) | $13.11 \pm 8.59\%$ | $(0.82\%, 36.78\%)$ | $98 \pm 2\%$ | $(92\%, 100\%)$ |

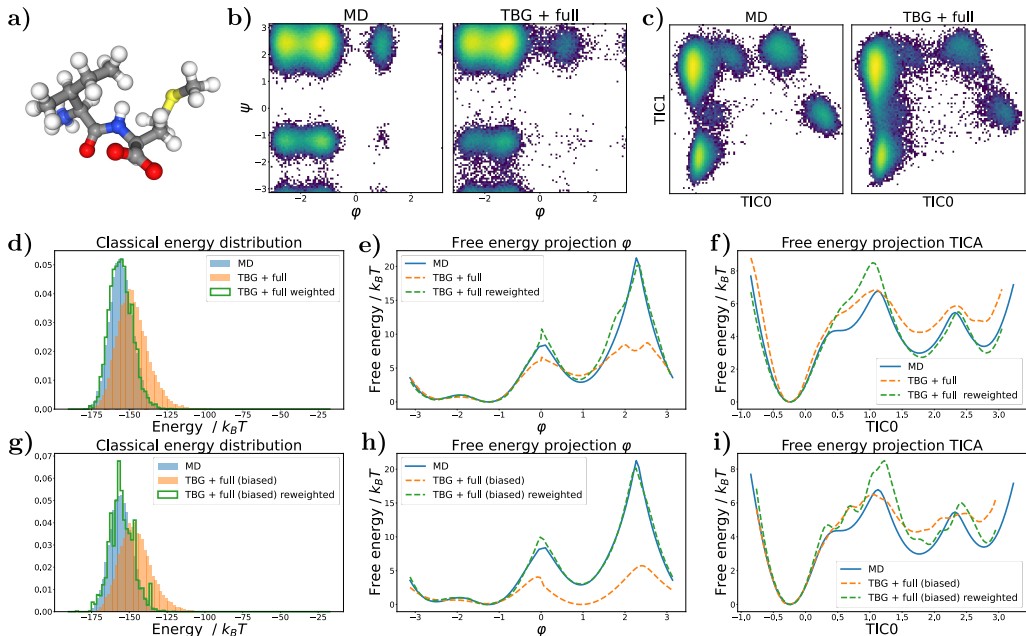

Figure 11: Results for the IM dipeptide (a) Sample generated with the TBG + full model (b) Ramachandran plot for the weighted MD distribution (left) and for samples generate with the TBG + full model (right). (c) TICA plot for the weighted MD distribution (left) and for samples generate with the TBG + full model (right). (d) Energies of samples generated with the TBG + full model. (e) Free energy projection along the $\varphi$ angle. (f) Free energy projection along the slowest transition (TIC0). (g) Energies of samples generated with the TBG + full (biased) model. (h) Free energy projection along the $\varphi$ angle for the TBG + full (biased) model. (i) Free energy projection along the slowest transition (TIC0) for the TBG + full (biased) model.

step and therefore also explore a potentially biased distribution rather than the unbiased Boltzmann distribution.

### A.7    Comparison with the Timewarp model [11]

Here, we provide additional results comparing our TBG + full model to the Timewarp model [11], as discussed in Section 5.3. In this second scenario, the objective is to explore all meta-stable states as given in the Ramachandran plot, aligning with the goals of a Boltzmann Emulator, where only approximate sampling from the Boltzmann distribution is required. For the TBG + full model, we do not apply reweighting, whereas in the Timewarp model, samples are rejected only if consecutive samples show an significant energy increase to prevent divergence [11].

Although reweighting is not strictly needed, we evaluate the energy of all samples generated by the TBG + full model, filtering out any rare, high-energy samples. We then compare both models based on the mean number of energy evaluations and the mean wall-clock time needed to explorae all states. As shown in Figure 21, the Timewarp model generally finds all states more quickly but requires more energy evaluations. Additionally, as illustrated in Figure 22, the Timewarp model's generated samples tend to exhibit higher energies compared to those from the TBG + full model. Consequently, the distribution produced by the TBG + full model more accurately approximates the target Boltzmann distribution.

For all Timewarp experiments, we used a proposal batch size of 100, as recommended in [11] for A-100 GPUs.

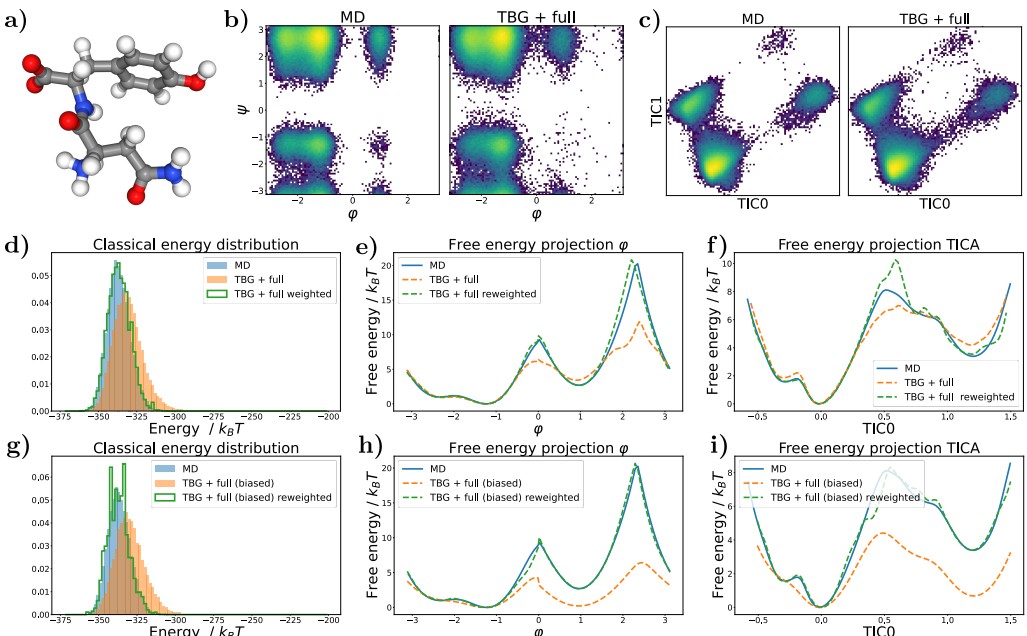

Figure 12: Results for the NY dipeptide (a) Sample generated with the TBG + full model (b) Ramachandran plot for the weighted MD distribution (left) and for samples generate with the TBG + full model (right). (c) TICA plot for the weighted MD distribution (left) and for samples generate with the TBG + full model (right). (d) Energies of samples generated with the TBG + full model. (e) Free energy projection along the $\varphi$ angle. (f) Free energy projection along the slowest transition (TIC0). (g) Energies of samples generated with the TBG + full (biased) model. (h) Free energy projection along the $\varphi$ angle for the TBG + full (biased) model. (i) Free energy projection along the slowest transition (TIC0) for the TBG + full (biased) model.

## B  Technical details

### B.1  Code libraries

We primarily use the following code libraries: *PyTorch* (BSD-3) [71], *bgflow* (MIT license) [8, 39], *torchdyn* (Apache License 2.0) [72], and *NetworkX* (BSD-3) [73] for validating graph isomorphisms. Additionally, we use the code from [41] (MIT license) for EGNNs, as well as the code from [11] (MIT license) and [23] (MIT license) for datasets and related evaluation methods.

Our code is available here: `https://osf.io/n8vz3/?view_only=1052300a21bd43c08f700016728aa96e`.

### B.2  Benchmark systems

The investigated benchmark systems were created in prior studies [23, 11].

**Alanine dipeptide**   The alanine dipeptide datasets were created in [23] (CC BY 4.0), we refer to them for detailed simulation details. The classical trajectory was created at $T = 300$K with the classical *Amber ff99SBildn* force-field. The subsequent relaxation was performed with the semi-empirical *GFN2-xTB* force-field [64].

**Dipeptides (2AA dataset)**   The original dipeptide dataset as introduced in [11] (MIT License) is available here: `https://huggingface.co/datasets/microsoft/timewarp`. As this includes a lot of intermediate saved states and quantities, like energies, we create a smaller version with is available here: `https://osf.io/n8vz3/?view_only=1052300a21bd43c08f700016728aa96e`. For a comprehensive overview of the simulation details, refer to [11]. All dipeptides were simulated

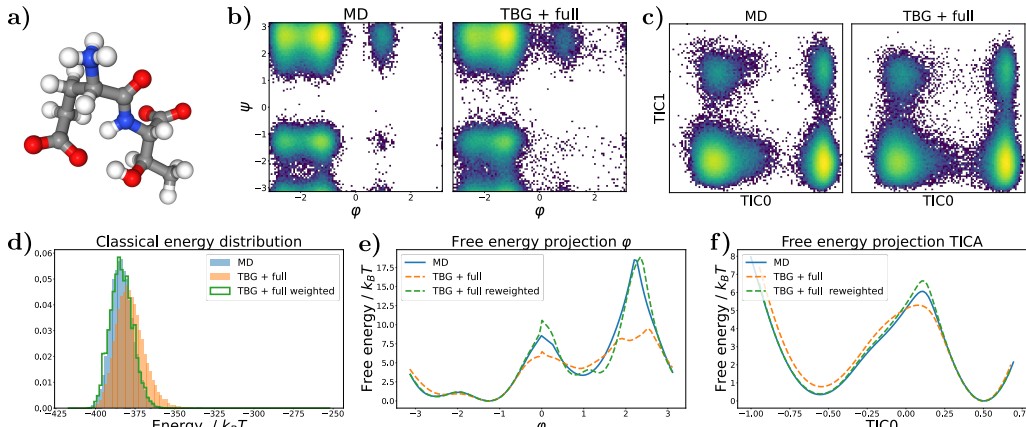

Figure 13: Results for the ET dipeptide (a) Sample generated with the TBG + full model (b) Ramachandran plot for the weighted MD distribution (left) and for samples generate with the TBG + full model (right). (c) TICA plot for the weighted MD distribution (left) and for samples generate with the TBG + full model (right). (d) Energies of generated samples (e) Free energy projection along the $\varphi$ angle. (f) Free energy projection along the slowest transition (TIC0).

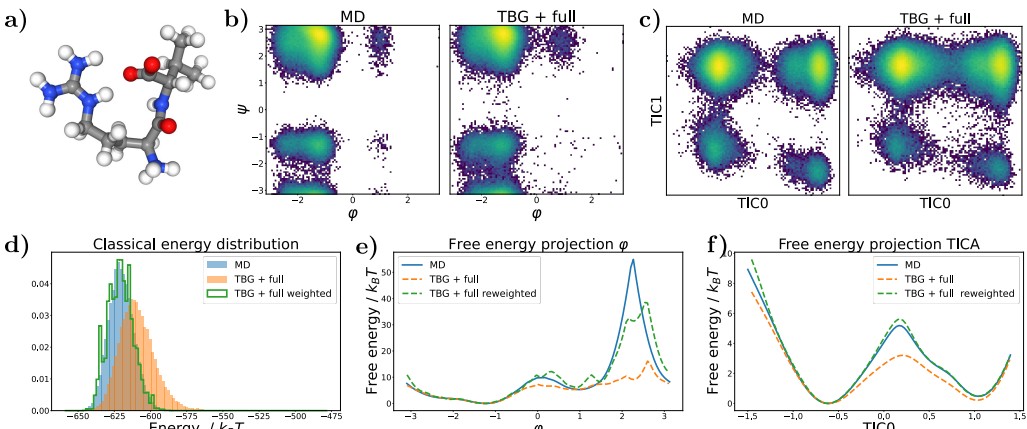

Figure 14: Results for the RV dipeptide (a) Sample generated with the TBG + full model (b) Ramachandran plot for the weighted MD distribution (left) and for samples generate with the TBG + full model (right). (c) TICA plot for the weighted MD distribution (left) and for samples generate with the TBG + full model (right). (d) Energies of generated samples (e) Free energy projection along the $\varphi$ angle. (f) Free energy projection along the slowest transition (TIC0).

with a classical *amber-14* force-field at $T = 310$K. The simulation of the training peptides were run for 50ns, while the test set peptides were run for $1\mu s$.

**Choice of test set peptides**   Inference is a costly process with CNFs (see Appendix B.7), and extensive sampling is necessary to obtain reliable estimates for the relative effective sample size (ESS). Therefore, we evaluate most of the transferable models on a subset of the test set. The dipeptides are randomly selected, but it is ensured that all amino acids are represented at least once. Only the best performing model (TBG + full) is evaluated on the whole test set of 100 dipeptides.

## B.3   Hyperparameters

We report the model hyperparameters for the different model architectures as describes in Section 4.1 in Table 5. As in [23] all neural networks $\phi_\alpha$ have one hidden layer with $n_{\text{hidden}}$ neurons and *SiLU* activation functions. The input size of the embedding $n_{\text{embedding}}$ depends on the model architecture.

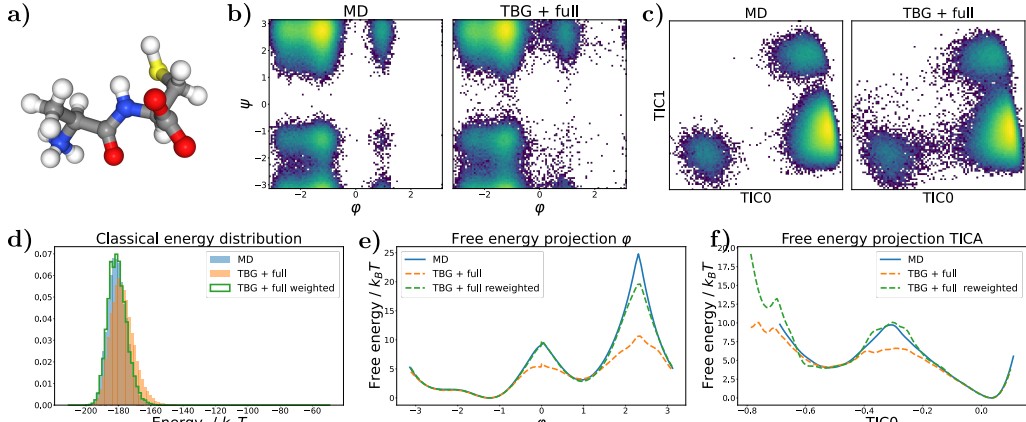

Figure 15: Results for the AC dipeptide (a) Sample generated with the TBG + full model (b) Ramachandran plot for the weighted MD distribution (left) and for samples generate with the TBG + full model (right). (c) TICA plot for the weighted MD distribution (left) and for samples generate with the TBG + full model (right). (d) Energies of generated samples (e) Free energy projection along the $\varphi$ angle. (f) Free energy projection along the slowest transition (TIC0).

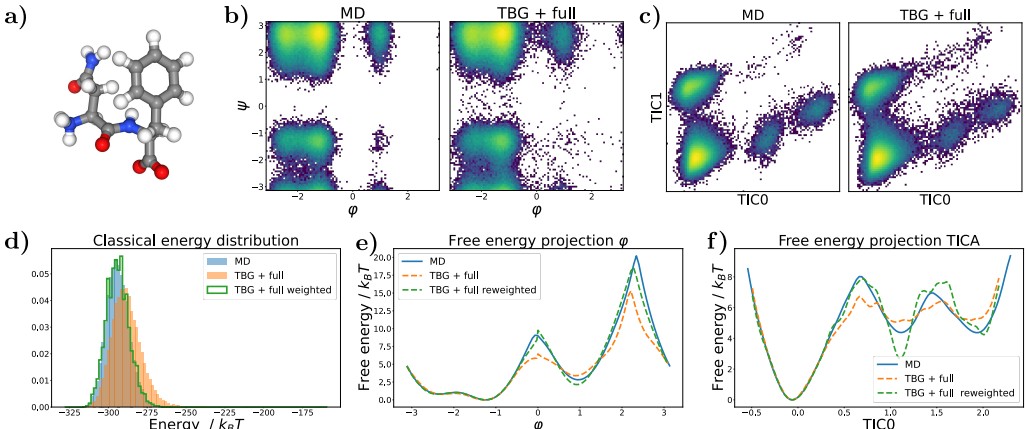

Figure 16: Results for the NF dipeptide (a) Sample generated with the TBG + full model (b) Ramachandran plot for the weighted MD distribution (left) and for samples generate with the TBG + full model (right). (c) TICA plot for the weighted MD distribution (left) and for samples generate with the TBG + full model (right). (d) Energies of generated samples (e) Free energy projection along the $\varphi$ angle. (f) Free energy projection along the slowest transition (TIC0).

We report training hyperparameters for the different model architectures in Table 6. It should be noted that all TBG models are trained in an identical manner if the training set is identical. We use the ADAM optimizer for all experiments [74]. For the dipeptide training, each batch consists of three samples for each peptide.

### B.4 Biasing target samples

As introduced in [23], it can be beneficial to bias the training data in such a way that unlikely states are more prominent. For alalnine dipeptide and many dipeptides, the positive $\varphi$ states at $\varphi = 1$ are often the unlikely ones and transition between the positive and negative $\varphi$ states are slow. For the alanine dipeptide dataset, the biasing methodology proposed in [23] is employed. Similarly, we bias the dipeptides based on the von Mises distribution $f_{\mathrm{vM}}$. The weights $\omega$ are computed along the $\varphi$ dihedral angle as

$$\omega(\varphi) = r \cdot f_{\mathrm{vM}} \left( \varphi | \mu = 1, \kappa = 10 \right) + 1, \tag{12}$$

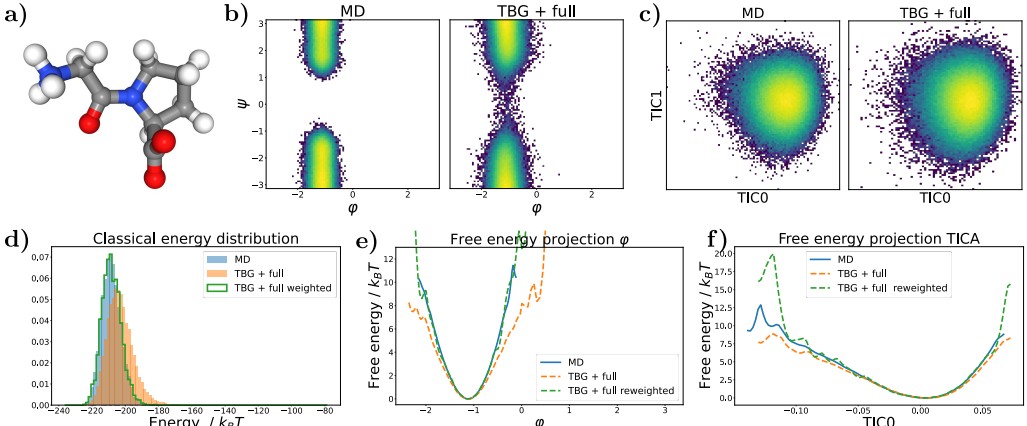

Figure 17: Results for the GP dipeptide (a) Sample generated with the TBG + full model (b) Ramachandran plot for the weighted MD distribution (left) and for samples generate with the TBG + full model (right). (c) TICA plot for the weighted MD distribution (left) and for samples generate with the TBG + full model (right). (d) Energies of generated samples (e) Free energy projection along the $\varphi$ angle. (f) Free energy projection along the slowest transition (TIC0).

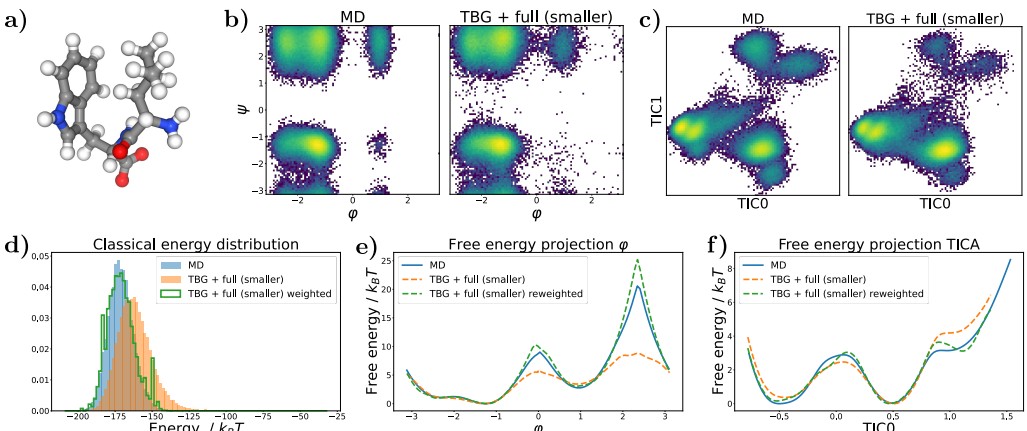

Figure 18: Results for the LW dipeptide for the TBG + full (smaller) model, which is trained on tenfold smaller trajectories than the TBG + full model. (a) Sample generated with the TBG + full (smaller) model (b) Ramachandran plot for the weighted MD distribution (left) and for samples generate with the TBG + full (smaller) model (right). (c) TICA plot for the weighted MD distribution (left) and for samples generate with the TBG + full (smaller) model (right). (d) Energies of generated samples. (e) Free energy projection along the $\varphi$ angle. (f) Free energy projection along the slowest transition (TIC0).

where $r$ is computed based on the ratio of positive and negative $\varphi$ states, such that both have nearly the same weight after the biasing.

## B.5 Encoding of atom types

The atom type embedding $a_i$ is a one-hot vector of $54$ classes. The classes are mostly defined by the atom type in the peptide topology. Therefore, only a few atoms are indistinguishable, such as hydrogen atoms that are bound to the same carbon or nitrogen atom. Moreover, we also treat oxygen atoms bound to the same carbon atom as indistinguishable, unless they are in the carboxyl group. Notably, we never treat particle groups as indistinguishable, such as two CH3 groups bound to the same carbon atom.

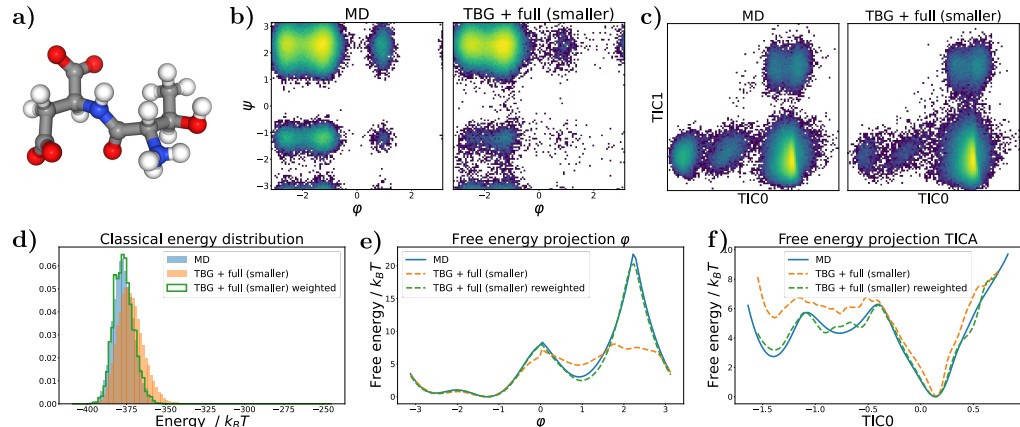

Figure 19: Results for the TD dipeptide for the TBG + full (smaller) model, which is trained on tenfold smaller trajectories than the TBG + full model. (a) Sample generated with the TBG + full (smaller) model (b) Ramachandran plot for the weighted MD distribution (left) and for samples generate with the TBG + full (smaller) model (right). (c) TICA plot for the weighted MD distribution (left) and for samples generate with the TBG + full (smaller) model (right). (d) Energies of generated samples (e) Free energy projection along the $\varphi$ angle. (f) Free energy projection along the slowest transition (TIC0).

Table 5: Model hyperparameters

| **Model** | $L$ | $n_{\text{hidden}}$ | $n_{\text{embedding}}$ | Num. of parameters |
|---|---|---|---|---|
| | | | alanine dipeptide | |
| BG + backbone | 5 | 64 | 8 | 147599 |
| TBG + full | 5 | 64 | 15 | 149147 |
| | | | Dipeptides (2AA) | |
| TBG | 9 | 128 | 5 | 1044239 |
| TBG + backbone | 9 | 128 | 13 | 1046295 |
| TBG + full | 9 | 128 | 76 | 1062486 |

## B.6 Effective samples sizes

The relative effective sample sizes (ESS) are computed with Kish's equation [56] as in prior work. For the alanine dipeptide experiments we use $2 \times 10^5$ samples for the forward ESS and $1 \times 10^4$ for the negative log likelihood computation. A total of $3 \times 10^4$ samples were used for each dipeptide in the forward ESS, while $4.5 \times 10^3$ samples were employed for the negative log likelihood computation.

Table 6: Training hyperparameters

| **Mdoel** | **Batch size** | **Learning rate** | **Epochs** | **Training time** |
|---|---|---|---|---|
| | | Alanine dipeptide | | |
| BG + backbone | 256 | $5e\text{-}4/5e\text{-}5$ | 500/500 | 3.5h |
| TBG + full | 256 | $5e\text{-}4/5e\text{-}5$ | 500/500 | 3.5h |
| | | Dipeptides (2AA) | | |
| TBG | 600 | $5e\text{-}4/5e\text{-}5/5e\text{-}6$ | 7/7/7 | 4d |
| TBG + backbone | 600 | $5e\text{-}4/5e\text{-}5/5e\text{-}6$ | 7/7/7 | 4d |
| TBG + full | 600 | $5e\text{-}4/5e\text{-}5/5e\text{-}6$ | 7/7/7 | 4d |
| TBG + full (smaller) | 600 | $5e\text{-}4/5e\text{-}5/5e\text{-}6$ | 30/30/30 | 2.5d |

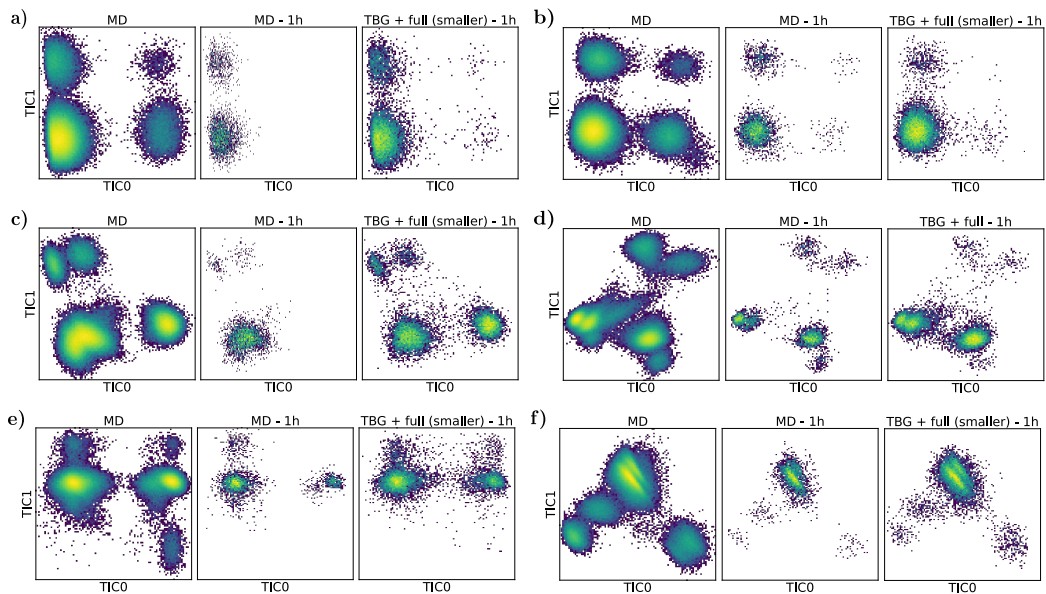

Figure 20: Comparison of classical MD runs for 1 hour (MD - 1h) and the sampling with the TBG + full (smaller) model without weight computation for 1 hour (TBG + full (smaller) - 1h). The TICA plots of different peptides from the test set are shown. It is important to note that the TICA projection is always computed with respect to the long MD trajectory (MD). All peptides stem from the test set. (a) CS dipeptide (b) EK dipeptide (c) KI dipeptide (d) LW dipeptide (e) RL dipeptide (f) TF dipeptide.

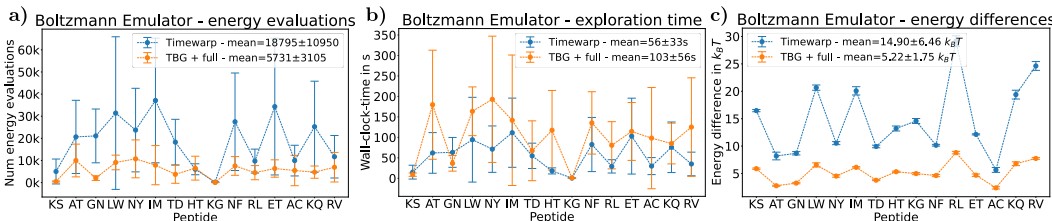

Figure 21: Boltzmann Emulator / Timewarp exploration experiments. Errors over 5 runs. Lower is better. (a) Number of energy evaluations until all states are found. (b) Wall-clock-time until all states are found. (c) Energy difference between the mean MD energy and the mean sample energy generated by the two methods.

## B.7 Computing resources

All training and inference was performed on single *NVIDIA A100 GPUs* with 80GB of RAM.

The training time for the models is reported in Appendix B.3, although it should be noted that a significant amount of time was required for hyperparameter tuning. It is estimated that at least ten times the compute time reported in Appendix B.3 was necessary to identify suitable hyperparameters. Furthermore, inference with CNFs is expensive, especially if one requires the reweighting weights. Generating $3 \times 10^4$ samples with the large transferable models for the dipeptides requires approximately four days, whereas generating $2 \times 10^5$ samples for the alanine dipeptide experiments takes less than one day. However, generating samples without corresponding weights significantly accelerates the sampling process. In the case of the dipeptides, the generation of $2 \times 10^5$ samples can be completed in less than one day. However, it should be noted that sampling can be done fully in parallel, as Boltzmann Generators generate independent samples.

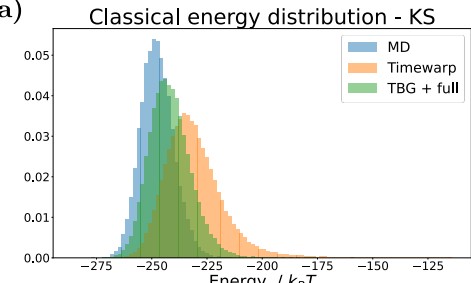
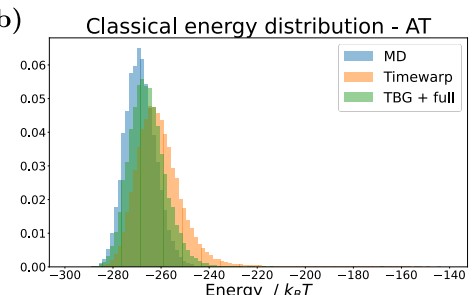

Figure 22: Comparison of energies of generated samples with a long MD simulation, Timewarp exploration, and TBG + full without reweighting. The energy distribution of the TBG + full model matches the Boltzmann distribution generated with MD better. (a) For the dipeptide KS. (b) For the dipeptide AT.

