# OpenReview forum: "Transferable Boltzmann Generators"
_NeurIPS.cc/2024/Conference — NeurIPS 2024 poster_

### Official Review · Reviewer_GrZN · 2024-07-08

**Soundness:** 3
**Presentation:** 3
**Contribution:** 3
**Rating:** 6
**Confidence:** 2

**Summary:**

The article "Transferable Boltzmann Generators" deals with the elaboration of a generative model based on Boltzmann Generators in order to estimate Boltzmann Distributions of molecules on which it has not been trained. The training procedure is based on continuous-time normalizing flow, where the trained model is then "transferred" to unseen molecules. Numerical experiments on dipeptides are provided to show how their method works.
The authors conclude that transferable Boltzmann Generators can be trained on some cases, when trained using normalizing flows.

**Strengths:**

The method introduced in this article is strongly evaluated on peptides. They illustrate with details how the samples generated by their methods (TBG+full or TBG+full reweighted) can work well (the latter model being better) and reproduce accurately the Ramachandran plots and the free energy projections.

The work provide a full set of details of the numerical implementation (number of parameters, batch size, learning rate, etc) in the appendices and many different experiments. In addition, their code is available online.

**Weaknesses:**

It is not clear how different their approach is from ref[22]. Maybe the authors could comment more on that.

**Questions:**

Can the authors comment on experiments on other types of data ?

**Limitations:**

The limitations are discussed adequately in the article.

---

> ### Author Rebuttal · Authors · 2024-08-07
>
> We thank the reviewer for their review and questions.
>
> > It is not clear how different their approach is from [22]. Maybe the authors could comment more on that.
>
> We agree with the reviewer that this aspect of our work is crucial and should be more prominently highlighted.
>
> As noted by the reviewer, our work builds upon the research presented in [22]. A major difference is in how we encode molecular structures in our model. In [22], the encoding methods involve either using solely the atom type (TBG) or adding different encodings for each atom in the peptide backbone (TBG + backbone). Therefore, there are not many different encodings for the atoms, and many atoms share the same encoding for both models.
> In contrast, our proposed model assigns embeddings based on the positions of an atom in the peptide as well as the corresponding amino acid it belongs to. This results in different embeddings for nearly all atoms. This embedding makes the model more expressive, as the EGNN generates updates based on pair wise inputs. If input pairs share the same embeddings with other pairs, the update contributions follow the same function. This uniform treatment may not be ideal, as atoms of the same atom type in different regions of the molecule can behave differently; however, the model treats them the same if their embeddings are identical.
>
> Moreover, [22] does not explore transferability to unseen systems. Our experiments demonstrate that their proposed architecture fails to generate significant effective sample sizes (ESS) for unseen dipeptides, as detailed in Section 5.2. In contrast, our TBG + full model achieves significant ESS across the entire test set (see Section 5.2 and Figure R4cd in the PDF of the global rebuttal).
>
> Thus, we provide the first demonstration of a transferable Boltzmann Generator, which is highly relevant to the field of AI and Science. Additionally, we propose a framework for transferable Boltzmann Generators using continuous normalizing flows,  that can be used with different vector fields, e.g. different equivariant models, and also includes post-processing of generated samples. We hope this framework will facilitate future research in this area.
>
> We will incorporate these points into the final version of the paper.
>
> > Can the authors comment on experiments on other types of data ?
>
> Our method is specifically tailored for molecular systems, particularly peptides and proteins. However, it could be extended to other small molecules or peptides simulated with more expensive force fields, such as semi-empirical ones are even first principle quantum mechanic force fields.  In these scenarios, energy evaluations become a primary bottleneck. Boltzmann Generators are particularly advantageous in this context, as they require several orders of magnitude fewer energy evaluations compared to MD simulations or other iterative methods
>
> There is potential to apply our approach to other types of data that exhibit symmetries, such as intersection traffic data [70] and point clouds [71]. Transferable models could be beneficial in these contexts as well. However, the choice of embeddings would need to be adapted based on the specific system of interest. It is challenging to predict how well our method would perform in these scenarios or identify potential additional obstacles.
>
> Additionally, while we did not apply our model to a different dataset, we evaluated it for a different task—namely, as a Boltzmann Emulator. The results of this evaluation can be found in the global rebuttal.
>
> [22] Leon Klein, Andreas Krämer, and Frank Noe. Equivariant flow matching. In NeurIPS 2023.
>
> [70] Berend Zwartsenberg et al. Conditional permutation invariant flows. arXiv preprint arXiv:2206.09021, 2022.
>
> [71] Marin Biloš and Stephan Günnemann. Equivariant normalizing flows for point processes and sets, 2021.

---

> > ### Comment · Reviewer_GrZN · 2024-08-09
> >
> > I thank the authors for their precise answer which clarify the point that I have raised. I'm changing "contribution" from 2->3.

---

> > > ### Author Response · Authors · 2024-08-11
> > >
> > > We are pleased that the reviewer is satisfied with our response.

---

### Official Review · Reviewer_84N3 · 2024-07-09

**Soundness:** 3
**Presentation:** 3
**Contribution:** 2
**Rating:** 5
**Confidence:** 4

**Summary:**

The paper tackles the challenging and high-impact problem of sampling from the high-dimensional Boltzmann distribution of molecular systems. The proposed method allows to train a Boltzmann generator that is applicable to systems it has not been trained on as demonstrated for a dataset of different Dipeptides. The proposed method leverages flow matching, equivariant neural networks and reweighting to sample from the target Boltzmann distribution in an unbiased manner and without auto-correlation between generated samples.

**Strengths:**

1. The proposed method represents a significant improvement over the state-of-the-art Boltzmann generators given that prior approaches are only applicable to systems they have been trained on, limiting their applicability in practice.
2. The experiments are appropriate to support the claim of transferability. The ablation studies based on limited and biased training data are relevant to judge practical applicability of the proposed method.
3. The paper clearly highlights the difference between Boltzmann generators and emulators and the importance of the former to predict downstream quantities in a physically rigorous manner.

**Weaknesses:**

1. The main methodological innovation consists of including topology information in the embedding. Otherwise, standard methods such as flow matching and reweighting are used. This seems like an incremental improvement.
2. In order to achieve real-world impact, a Boltzmann generator needs to be applicable to large molecular systems, but the dataset of Dipeptides in gas phase is not representative of real-world applications. Given that the proposed method depends strongly on reweighting generated samples to the true Boltzmann distribution and reweighting efficiency decreases exponentially with the system size, scaling the approach to large system of practical relevance appears challenging.
3. The employed EGNN architecture is not considered state-of-the-art anymore and results with more recent models for the vector field would be interesting.
4. The authors did not compare their method to the recent TimeWarp method, which tackles the same problem (even using the same dataset), but leverages an existing initial structure to sample in an unbiased, but autocorrelated manner using Hamiltonian Monte Carlo. To judge the merits of the proposed method, it seems important to compare these two methods (using the same topological embedding scheme) in terms of sampling efficiency for a given computational budget and transferability.  In addition, the TimeWarp paper also benchmarks their method against Tetrapeptides. It would be interesting to see which of the two methods generalizes better to larger systems and how this impacts sampling efficiency.

**Questions:**

Did the authors compare the proposed method against the TimeWarp method?

**Limitations:**

The authors discuss the limitations of the proposed approach, including the limited size of molecular systems studied, as well as directions for further research, such as using more advanced flow matching approaches, informative prior distributions and more different model architectures.

---

> ### Author Rebuttal · Authors · 2024-08-07
>
> We thank the reviewer for their detailed review, insightful questions, and valuable suggestions. We will now address their comments and questions individually.
>
> > The main methodological innovation consists of including topology information in the embedding. Otherwise, standard methods such as flow matching and reweighting are used. This seems like an incremental improvement.
>
> We agree with the reviewer that our proposed method builds upon prior work on Boltzmann Generators and flow matching, particularly the work of [22]. However, we demonstrate for the first time that a transferable Boltzmann Generator capable of achieving relevant effective sample sizes is feasible. This capability was not possible with previous architectures, as evidenced by our experiments. Therefore, we do not consider the improvement to be incremental. Additionally, we introduce a framework for transferable Boltzmann Generators using continuous normalizing flows. This framework is adaptable to various architectures, including different equivariant models, and incorporates post-processing of generated samples. We hope this framework will facilitate further research in this area.
>
> > Real world impact of Boltzmann Generators and scalability.
>
> We agree with the reviewer that scaling Boltzmann Generators to larger systems remains a significant challenge. However, we see several potential pathways forward. Scaling to larger systems often involves coarse-graining, which typically results in the loss of an explicit energy function. In such cases, the transferable Boltzmann Generator would effectively become a transferable Boltzmann Emulator. Depending on the specific application, samples from a distribution close to the target Boltzmann distribution may be sufficient. Our results in the general rebuttal show that unweighted samples from the TBG + full model closely resemble the target Boltzmann distribution.
> It remains to be explored to what extent Boltzmann Generators can be scaled to larger systems.
> Additionally, small systems can still be highly relevant, especially when paired with more expensive force fields, such as semi-empirical or first-principles quantum-mechanical force fields. In these scenarios, energy evaluations become a primary bottleneck. Boltzmann Generators are particularly advantageous in this context, as they require several orders of magnitude fewer energy evaluations compared to MD simulations or other iterative methods, such as Timewarp [10]. One potential real world application would be the simulation of small ligands with expensive force fields.
>
> We will expand our discussion of these points in the limitations section of the final version.
>
> > The employed EGNN architecture is not considered state-of-the-art anymore and results with more recent models for the vector field would be interesting.
>
> We agree that there are more expressive EGNN architectures available. However, the one used in our work is particularly efficient to evaluate, which is crucial given the hundreds of vector field evaluations required for inference. Since our results are already quite promising—e.g., a median effective sample size (ESS) of over 10% on the test set (see Figure R4c) and nearly all generated samples have correct configurations (see Figure R4d)—we have focused on this efficient architecture. Experiments with different vector field architectures are left for future research, but we have provided a framework to facilitate such investigations.
>
> > Comparison to Timewarp [10] for different computational budgets
>
> We thank the reviewer for this suggestion and agree that it is an important method to compare against. We present the results in the global rebuttal, where we demonstrate that our TBG + full model outperforms Timewarp, particularly in terms of energy evaluations, which are crucial as discussed above.
>
> We use the Wasserstein distance between generated distributions as our evaluation metric rather than effective sample size (ESS). The reported ESS for Timewarp is based on autocorrelations of a specific collected variable, which contrasts with Kish's ESS used for Boltzmann Generators. This discrepancy makes direct comparison of ESS values challenging.
>
> Timewarp already uses a similar embedding scheme as our proposed methods, as it takes the current position and types of atoms as an input.
> Therefore, Timewarp already has all the information about the bond graph.
> We concur with the reviewer that experiments with tetrapeptides would be valuable. However, due to the significant computational resources required for such larger experiments, we did not conduct them. We hope future research will explore these systems
>
> [10] Leon Klein et al. Timewarp: Transferable acceleration of molecular dynamics by learning time-coarsened dynamics. In NeurIPS 2023.
>
> [22] Leon Klein, Andreas Krämer, and Frank Noe. Equivariant flow matching. In NeurIPS 2023.

---

> > ### Comment · Reviewer_84N3 · 2024-08-08
> >
> > The provided comparisons to the TimeWarp method clearly improve the paper and underline the merit of TBGs over existing approaches.
> >
> > In particular, the reasoning behind the different embedding requirements for Boltzmann Generators vs methods that have access to an initial structure (such as TimeWarp), is noteworthy and should be included in a more thorough discussion of the embeddings in the paper.

---

> > > ### Author Response · Authors · 2024-08-11
> > >
> > > We would like to thank the reviewer once again for suggesting the Timewarp experiments. We will include the results, along with the suggested discussion of the embedding, in the final version of the paper.

---

### Official Review · Reviewer_AbVU · 2024-07-11

**Soundness:** 3
**Presentation:** 3
**Contribution:** 3
**Rating:** 6
**Confidence:** 2

**Summary:**

The authors proposes Transferable Boltzmann Generators (TBGs), which are transferrable for approximating the target distribution of unseen molecular datasets. TBGs are based on the graph-based continuous normalizing flow, and trained by using the simple flow matching. The authors experimentally demonstrate that the proposed model outperforms its competitors.

**Strengths:**

The motivation of the paper is very clear and relevant to the field of the AI + Science. The experimental results seem to be convincing, supported by thorough ablation studies (though it seems there is a lack of comparison with some previous works, please see Weaknesses).

**Weaknesses:**

I think the authors should put a more effort on explaining why the proposed TBGs are more transferrable than the previous models, for examples, coordinates-based Boltzmann generators [1]. I found that the proposed architecture is heavily based on [1], and the clear difference between [1] and the proposed one is the use of the auxiliary inputs $b_i$ and $c_i$ (and possibly the modification of $a_i$, though I am not sure what is “the topology for a classical force field”, which seems to be the added information compared to the previous $a_i$ of [22], i.e., “simply the atom types of the backbone encoding”). In the current version of this paper, the description about the proposed framework is only approximately one-page, and it is not unclear why the model can be transferred to the untrained dataset. To hedge this, the authors might explain the clear difference between BGs and TBGs, with emphasizing the usefulness of the incorporated inputs and modifications, with particular consideration for readers who are not familiar with physical chemistry.

***

In Section 2, the authors introduce the Boltzmann Emulators, which are transferable due to removing the constraint of weighted sampling [2, 3]. However, the authors do not compare the proposed TBGs with these methods for the transferability experiment. I understand the Boltzmann Emulators lack the unbiased estimations, but I believe the experimental evidence should be provided for the completeness of the authors’ claim.

***

[1] Leon Klein, Andreas Krämer, and Frank Noe. Equivariant flow matching. In Thirty-seventh Conference on Neural Information Processing Systems, 2023.

[2] Osama Abdin and Philip M Kim. Pepflow: direct conformational sampling from peptide energy landscapes through hypernetwork-conditioned diffusion. bioRxiv, pages 2023–06, 2023.

[3] Juan Viguera Diez, Sara Romeo Atance, Ola Engkvist, and Simon Olsson. Generation of conformational ensembles of small molecules via surrogate model-assisted molecular dynamics. Machine Learning: Science and Technology, 5(2):025010, 2024.

**Questions:**

Please see Weaknesses.

**Limitations:**

The authors adequately address the limitation, including the lack of evaluation on large-scale systems, and future directions, e.g., the use of other training objectives and prior distributions, for this work.

---

> ### Author Rebuttal · Authors · 2024-08-07
>
> We thank the reviewer for their review and questions. We now address their comments individually.
>
> > I think the authors should put a more effort on explaining why the proposed TBGs are more transferrable than the previous models, for examples, coordinates-based Boltzmann generators (...) The authors should explain the clear difference between BGs and TBGs, with emphasizing the usefulness of the incorporated inputs and modifications, with particular consideration for readers who are not familiar with physical chemistry.
>
> We agree with the reviewer that this aspect of our work is crucial and should be more prominently highlighted and presented in a way that is accessible to readers less familiar with physical chemistry.
>
> As noted by the reviewer, our work builds upon the research presented in [1]. A major difference is in how we encode molecular structures in our model. In [1], the encoding methods involve either using solely the atom type (TBG) or adding different encodings for each atom in the peptide backbone (TBG + backbone).  The backbone of a peptide consists of a repeating sequence of atoms, which form the sequence. Each amino acids contributes the same atoms to the backbone. Therefore, there are not many different encodings for the atoms, and many atoms share the same encoding for both models.
> In contrast, our proposed model assigns embeddings based on the positions of an atom in the peptide as well as the corresponding amino acid it belongs to. This results in different embeddings for nearly all atoms. This embedding makes the model more expressive, as the EGNN generates updates based on pair wise inputs. If input pairs share the same embeddings with other pairs, the update contributions follow the same function.  This uniform treatment may not be ideal, as atoms of the same atom type in different regions of the molecule can behave differently; however, the model treats them the same if their embeddings are identical.
>
> Moreover, [1] does not explore transferability to unseen systems. Our experiments demonstrate that their proposed architecture fails to generate significant effective sample sizes (ESS) for unseen dipeptides, as detailed in Section 5.2. In contrast, our TBG + full model achieves significant ESS across the entire test set (see Section 5.2 and Figure R4cd in the PDF of the global rebuttal).
>
> Thus, we provide the first demonstration of a transferable Boltzmann Generator, which is highly relevant to the field of AI and Science, as noted by the reviewer. Additionally, we propose a framework for transferable Boltzmann Generators using continuous normalizing flows,  that can be used with different vector fields, e.g. different equivariant models, and also includes post-processing of generated samples. We hope this framework will facilitate future research in this area.
>
> We will incorporate these points into the final version of the paper.
>
> > Comparison with Boltzmann Emulators
>
> We thank the reviewer for the suggestion and agree that this comparison is valuable. We have conducted additional experiments using the TBG + full model as a Boltzmann Emulator and compared it to Timewarp [10], which is also suitable as a transferable Boltzmann Emulator for small peptide systems. We selected Timewarp over [2] and [3] because [2] is designed for larger systems, and [3] is not currently applicable to molecules with rings, which are common in many of the dipeptides studied in our work.
>
> Additionally, we compared the TBG + full model used as a Boltzmann Generator with the Timewarp model combined with Metropolis-Hastings. Our findings demonstrate that the TBG + full model performs better in both scenarios. Please refer to the global rebuttal for detailed results.
>
> [10] Leon Klein et al. Timewarp: Transferable acceleration of molecular dynamics by learning time-coarsened dynamics. In NeurIPS 2023.

---

> > ### Comment · Reviewer_AbVU · 2024-08-12
> >
> > Thank you for the authors' thoughtful response, especially the newly conducted comparison with Timewarp. Most of my concerns have been resolved, so I will be raising the review score.

---

> > > ### Author Response · Authors · 2024-08-14
> > >
> > > We are happy that we could address the reviewer's concerns and would like to thank them again for suggesting the additional experiments for Boltzmann Emulators.

---

### Official Review · Reviewer_C3Gj · 2024-07-13

**Soundness:** 3
**Presentation:** 3
**Contribution:** 3
**Rating:** 7
**Confidence:** 4

**Summary:**

- This work builds upon Equivariant flow matching (Klein, 2023) and proposes a transferable Boltzmann Generator that sample Boltzmann distribution for molecules outside the training set. It is a common scenario that simulation data are often scarce for the system of interest and model transferability is highly desired.
- The key contribution of this work is expanding an existing system-dependent framework to the transferable setting, and pinpointing the importance of proper topology encoding in previous models.
- Together with the advantages of normalizing flows trained with score matching, the authors showed the model have superb iid sampling capability of Boltzmann distributions on the model system of alanine dipeptide (single protein) and strong transferability between different dipeptides.
- Through comprehensive analyses, the authors showed that proposed model is accurate, data efficient, and has potential to sample unseen metastable states.

**Strengths:**

- While being an followup work on Equivariant flow matching (Klein, 2023), this work pioneered in providing the first proof-of-the-concept work that Boltzmann Generators learned through flow matching has potential transferability to unseen systems.
- During this exploration, they note the importance of encoding topology information through atom encoding to avoid indistinguishable atoms and transferability to unseen molecules.
- Additionally, they conducted comprehensive analysis and ablation studies to answer key questions on models ability of 1) recovering the exact Boltzmann distribution and free energies along the reaction coordinates, 2) sample unknown metastable states, and 3) the influence of training on limited data. These information are valuable for communities to better understand model’s capability.
- Overall, it is a pioneering, valuable, and well-written work that improving the solution towards an important question in AI for science.

**Weaknesses:**

The main weakness is the lack of experiments on the transferability in multiple systems. The authors only demonstrated the transferability using a small system of dipeptides, which has limited complexities. Despite being a proof-of-the-concept work, more efforts on improving model scalability and verifying on larger transferable systems would have improved the contribution and significance of this work.

**Questions:**

1. Can the authors clarify the differences between three “architectures” of TBG-full, TBG-backbone, and TBG? This question remains throughout the reading and I current understand that the main difference is the atom encoding (i.e., topology informations) while all other model specifications are remain the same. Specifically:
    - TBG+full: atom 54 encoding, unique encoding for most atoms.
    - TBG/BG+backbone: unique encoding for backbone atoms + atom element types for side chain atoms.
    - TBG: 5 atom element types

    If so, can the authors elaborate on the reason of the performance difference (albeit small) between BG-Backbone and TBG-full in Alanine dipeptide experiment?


1. As the isomorphism problem (chemical bonding and permutation symmetry) can be largely mitigated by unique atom encoding, the chirality remains a problem. While the authors uses a post-processing to filter valid molecules, one might wonder if the treatment scales as the system becomes larger and includes more chiral centers. Is there any other means to avoid chirality issues?

1. Line 234 - 235: the authors state that the performance for the classical force field is much better than the semi-empirical one because the training data stems from the target distribution. While data was simulated using a classical force field AND subsequently relaxed with respect to the semi-empirical force field; and the objectives if to model the Boltzmann distribution defined by the semi-empirical force field (line 216 - 219). Can the authors help to better understand which is the “target distribution” and which distribution does the training data follow?

1. Minor typo: eq 11 in the summation i → j for removing the center of mass

**Limitations:**

The authors extensively discussed the limitations and further justifications of current work, including 1) scaling to larger systems; 2) different flow matching forms; 3) different priors; 4) training peptides diversity; and 5) alternative architectures with enhanced performance.

---

> ### Author Rebuttal · Authors · 2024-08-07
>
> We thank the reviewer for their detailed review and questions. We now address their comments individually. To keep the rebuttal within the character limit, we often only cite parts of each question.
>
> > Lack of experiments on the transferability in multiple systems and scalability
>
> We acknowledge the reviewer's point that testing our methods on larger systems or systems of varying sizes would be valuable. However, due to the significant computational resources required, we did not conduct these experiments. We hope that future research will explore these areas and that our proposed TBG framework will prove useful in such investigations.
>
> > Can the authors clarify the differences between the three “architectures” of TBG-full, TBG-backbone, and TBG?
>
> The reviewer describes the differences mostly correctly. The TBG-full architecture includes not only encoding for the atom type but also for the amino acid to which the atom belongs. This additional encoding allows the TBG-full model to perform better on alanine dipeptide and other dipeptides, as it can better differentiate between atom pairs and is, therefore, more expressive. The EGNN generates updates based on pairwise inputs. If input pairs share the same embeddings as other pairs, the update contributions follow the same function. This uniform treatment may not be ideal, as atoms in different parts of the molecules can behave differently, yet the model must treat them the same if their embeddings are identical.
>
> > Architecture that respects chirality
>
> We are unsure if this is feasible, as we start from Gaussian noise rather than from a configuration with the correct chirality. Therefore, a chirality-conserving architecture would not be beneficial. However, it is important to note that even the current TBG+full model can distinguish between different chiralities because the distances between atoms vary based on the chirality. The model is only equivariant with respect to global reflections, which we do not consider as representing different chiralities, since they can be resolved by mirroring.
>
> > Classical vs semi-empirical force field performance
>
> We conducted two experiments with alanine dipeptide. In the first experiment, the target is the Boltzmann distribution defined by the classical force field. Here, the training data originates from an MD trajectory run with the classical force field, so the training distribution matches the target distribution.
>
> In the second experiment, the target is the Boltzmann distribution specified by the semi-empirical force field. However, the training data was not generated from a long MD simulation with the semi-empirical force field. Instead, it was obtained by relaxing samples from the classical MD simulation for a few steps with respect to the semi-empirical force field. As a result, these training samples do not stem from the Boltzmann distribution defined by the semi-empirical force field. This discrepancy is evident from the free energy differences between the positive and negative phi states, as shown in Table 2 of [22], where the free energy difference in the training data significantly differs from the value obtained through Umbrella sampling with the semi-empirical force field.
>
> We acknowledge that this may not have been clearly communicated in the current version of the paper and will provide a more detailed explanation in the final version.
>
> > Typo in eq. 11
>
> Thanks! That is a good catch.
>
> [22] Leon Klein, Andreas Krämer, and Frank Noe. Equivariant flow matching. In NeurIPS 2023.

---

> > ### Comment · Reviewer_C3Gj · 2024-08-11
> >
> > I thank the authors for the clarification on the points raised - my questions have been resolved and happy to maintain the original rating.

---

> > > ### Author Response · Authors · 2024-08-14
> > >
> > > We are glad that we could address all the reviewer's questions.

---

### Author Rebuttal · Authors · 2024-08-06

We appreciate the reviewers for their time and insightful feedback on our paper.

In response to the reviewers' request, we have included a comparison of our Transferable Boltzmann Generator (TBG) with the Timewarp model [10]. Unlike our approach, which generates independent samples, the Timewarp model learns to predict large time steps, which are combined with Metropolis-Hastings acceptance steps to ensure asymptotically unbiased samples. For this comparison, we used the publicly available weights for the Timewarp model, which was trained on the same dipeptide dataset we employed in our experiments. Notably, the Timewarp model required nearly three weeks of training on four A-100 GPUs, representing a training budget approximately 30 times greater than that of the TBG + full model.

We conducted a comparison of the two methods using 16 peptides from the test set. The corresponding figures can be found in the attached PDF.

1. In the first scenario, the goal is to generate samples from the target Boltzmann distribution for dipeptides unseen during training.     Timewarp accomplishes this by iteratively proposing samples and accepting or rejecting them using the Metropolis-Hastings algorithm, resulting in an asymptotically unbiased Markov chain. This approach aligns with the typical objective of a Boltzmann Generator.
We generated samples using both methods under fixed computational budgets and evaluated the results by comparing the Wasserstein distance between the generated Ramachandran plot and one generated from a long MD simulation. The computational budgets were: (a) 30,000 energy evaluations, (b) 12 hours, and (c) 24 hours of wall-clock simulation time on an A-100 GPU. As shown in Figure R1, the TBG + full model outperformed the Timewarp model across all budgets, particularly in terms of energy evaluations.
Energy evaluations are especially critical, as they become a major computational bottleneck with more complex force fields, such as semi-empirical or even quantum mechanical force fields. Furthermore, we present a comparison of the decay of the Wasserstein distance in Figure R2a ad R2b.
Notably, Timewarp occasionally fails to explore all states within the 24-hour computational budget, as shown in Figure R2d, whereas the TBG + full model consistently identifies all states within the same budget (see Figures in the main paper and appendix). We still compute the Wasserstein distance in these cases.

2. In the second scenario, the objective is to explore the most unlikely state, indicated by an orange circle in Figure R2d, which is the most unlikely state for most dipeptides. This goal aligns with that of a Boltzmann Emulator, as only approximate sampling from the Boltzmann distribution is required. For the TBG + full model, we do not apply reweighting, while for the Timewarp model, generated samples are only rejected if the energy increases by more than 300 $k_B T$ in consecutive samples, which is essential to prevent divergence [10]. Although not strictly necessary, we evaluated the energy of all samples generated with the TBG + full model and filtered out high-energy samples.
We compared the mean number of energy evaluations and the mean wall-clock time required to discover the state. Timewarp generally finds the state more quickly but requires more energy evaluations, as shown in Figure R3a and R3b. Additionally, the energies of samples generated with the Timewarp model tend to be higher than those generated with the TBG + full model, as illustrated in Figure R3c and R4a and R4b. Therefore, the distribution generated by the TBG + full model more closely approximates the target Boltzmann distribution.

For all Timewarp experiments, we used a proposal batch size of 100, as recommended in [10] for A-100 GPUs.

We will include these experiments in the final version of the paper.

[10] Leon Klein et al. Timewarp: Transferable acceleration of molecular dynamics by learning time-coarsened dynamics. In NeurIPS 2023.

---

### Decision · Program_Chairs · 2024-09-25

**Decision:**

Accept (poster)

**Comment:**

The article presents a generative method for equilibrium samples of molecular systems that allows approximate sampling from the target distribution of unseen systems.

Reviewers indicate the article deals with a challenging and high impact problem and expands an existing system-dependent framework to the transferable setting. They find that it provides an important proof of concept, presents comprehensive analysis and ablation studies, and provides a full set of details on the numerical implementation. They praised it as a pioneering, valuable, and well-written work towards an important question in AI for science, and a significant improvement over the state of the art. Some weaknesses were listed including on the methodology, comparisons, and scalability. The rebuttal offered a comparison with Timewarp that was suggested by reviewers and resolved most concerns in initial reviews, prompting raised review scores.

In view of the positive reception of the article, I am recommending accept. Authors are asked to implement the promised additions in the final version of the article, particularly discussion of embedding and Timewarp experiments.